# scMoMaT jointly performs single cell mosaic integration and multi-modal bio-marker detection

Ziqi Zhang [1], Haoran Sun[2], Ragunathan Mariappan[3], Xi Chen [4], Xinyu Chen[5], Mika S. Jain[6], Mirjana Efremova [7], Sarah A. Teichmann [6], Vaibhav Rajan[3] & Xiuwei Zhang [1] ✉

Single cell data integration methods aim to integrate cells across data batches and modalities, and data integration tasks can be categorized into horizontal, vertical, diagonal, and mosaic integration, where mosaic integration is the most general and challenging case with few methods developed. We propose scMoMaT, a method that is able to integrate single cell multi-omics data under the mosaic integration scenario using matrix tri-factorization. During integration, scMoMaT is also able to uncover the cluster specific bio-markers across modalities. These multi-modal bio-markers are used to interpret and annotate the clusters to cell types. Moreover, scMoMaT can integrate cell batches with unequal cell type compositions. Applying scMoMaT to multiple real and simulated datasets demonstrated these features of scMoMaT and showed that scMoMaT has superior performance compared to existing methods. Specifically, we show that integrated cell embedding combined with learned bio-markers lead to cell type annotations of higher quality or resolution compared to their original annotations.

The advance in single cell multi-omics technology makes it possible to profile a single cell from multiple modalities. Single cell RNA-sequencing (scRNA-seq) is able to measure the gene expression of individual cells, whereas single cell ATAC-sequencing (scATAC-seq) measures the chromatin accessibility of individual cells. On the other hand, new sequencing technologies have been proposed to profile more than one modality in a cell simultaneously. There exist technologies that are able to profile both protein abundance and gene expression[1], chromatin accessibility and gene expression[2], or chromatin accessibility and protein abundance[3] within a cell at the same time. Integrating cells from multiple modalities provides a comprehensive view of cellular identity and the key features (e.g. chromatin regions, genes, proteins, etc) that define the identity, and can further help to understand the underlying cross-modalities relationships.

Data integration tasks on such single cell data matrices can be categorized into four different scenarios[4]: horizontal integration, or termed batch effect removal, refers to the case where all data batches have the same modality. Vertical integration refers to the case where a data batch is measured with multiple modalities. Diagonal integration refers to the case that neither cells nor modalities are shared between data matrices. Mosaic integration is the most general case and can be any combination of horizontal, vertical, and diagonal integration. Considering an $m \times b$ grid that corresponds to $m$ modalities and $b$ batches, mosaic integration methods aim to integrate any subset of data matrices from this grid.

Various methods have been proposed to deal with these integration scenarios[4]. LIGER[5] and Seurat v3[6] were developed for horizontal and diagonal integration tasks. CoupleNMF[7], MMD-MA[8],

---

[1]School of Computational Science and Engineering, Georgia Institute of Technology, Atlanta, GA, USA. [2]School of Mathematics, Georgia Institute of Technology, Atlanta, GA, USA. [3]Department of Information Systems and Analytics, National University of Singapore, Singapore, Singapore. [4]Department of Biology, Southern University of Science and Technology, Shenzhen, Guangdong, China. [5]Bioengineering Program, Georgia Institute of Technology, Atlanta, GA, USA. [6]Wellcome Sanger Institute, Hinxton, UK. [7]Cancer Research UK Barts Center, London, UK. ✉e-mail: xiuwei.zhang@gatech.edu

scDART[9] were developed for diagonal integration task. Seurat v4[10], scAI[11], and MultiVI[12] were developed for vertical integration task. Recently, new methods have been proposed to work with less restricted integration scenarios. Bridge integration[13] uses one jointly profiled data batch that included all modalities as the "bridges" and integrates all data batches using dictionary learning. Cobolt[14] employs a multimodel variational autoencoder framework to learn the cell representations. Both methods require one batch of cells where all modalities were measured. Recently proposed methods, including MultiMap[15], UINMF[16] and StabMap[17], can integrate data matrices in mosaic integration scenario.

When integrating data matrices from multiple batches and one modality, the goal is to learn cell representations where batch effects are removed and cell identities are preserved. When integrating data from multiple batches and multiple modalities, the multi-modality data should yield more output than single-modality data. However, existing mosaic integration methods still focus on learning cell representations, though the use of multi-modality data was shown to lead to better cell embedding in terms of certain metrics compared to single-modality data[16].

Here we propose scMoMaT (single cell Multi-omics integration using Matrix Tri-factorization), a data integration framework that is designed to integrate an arbitrary number of data matrices under mosaic integration scenario (Fig. 1a). Apart from integrating cells, scMoMaT aims to exploit the multi-modality data: scMoMaT learns cell type specific bio-markers across modalities, including marker genes (from the gene expression modality), marker motifs or regions (from the chromatin accessibility modality) and marker proteins (from the protein abundance modality). It extracts the bio-marker not only from

the original features of the data matrices, but also from the features that are generated by other methods. For example, users can add motif deviation matrices (learned from the original scATAC-seq matrix through chromVAR[18], representing the motif activities within cells) to the input, and scMoMaT can learn the motif markers in addition to the bio-markers from the original modalities. These bio-markers can be used to interpret the cell clusters and annotate cell types with evidence from multiple modalities. In addition, scMoMaT does not assume cells to have similar distribution across batches and can integrate cell batches with disproportionate cell type compositions. We test scMoMaT on both real and simulated datasets covering various kinds of integration tasks. We first test scMoMaT on multiple simulated datasets and quantitatively evaluate its performance. We then test scMoMaT on four real datasets covering different integration scenarios, including one human PBMC dataset, one mouse brain cortex dataset, one human bone marrow dataset, and one mouse spleen dataset. We compare the performance of scMoMaT with state-of-the-art data integration methods using multiple benchmarking metrics. The results show that scMoMaT has superior performance in learning cell embedding and dealing with disproportionate cell type composition between batches. We demonstrate how the multi-modal bio-markers we learned can be used to annotate cell types of the clusters obtained in the integrated space. We also show that these annotations can be better than the annotations published together with the dataset.

## Results

### Framework of scMoMaT

scMoMaT uses a matrix tri-factorization framework, which treats each single cell data matrix as a relationship matrix between the "cell" and

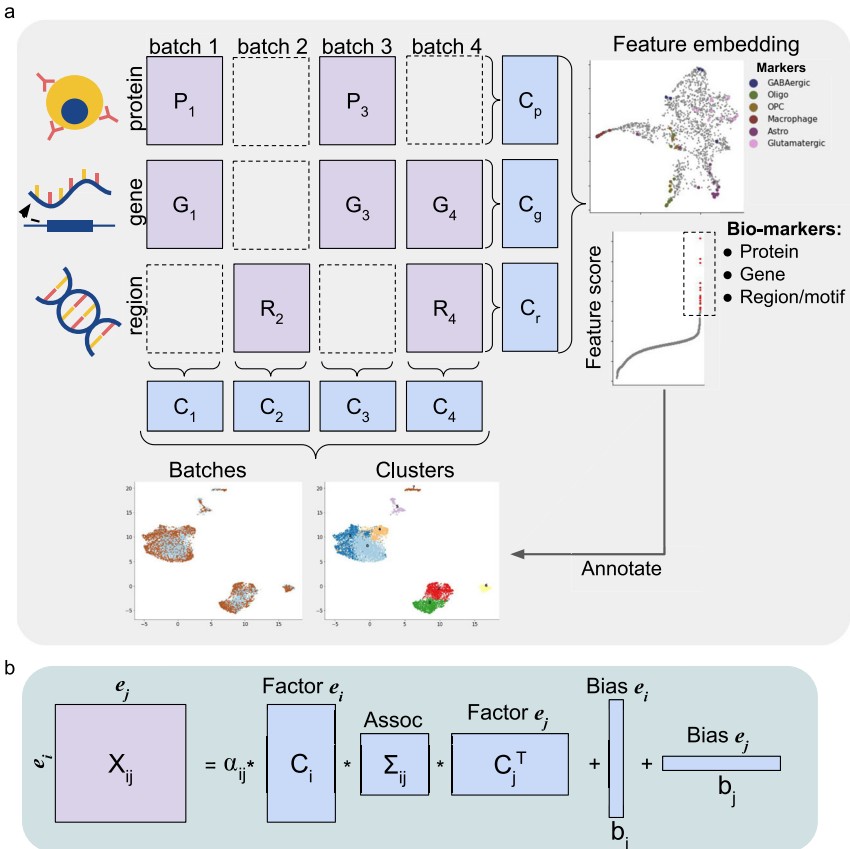

**Fig. 1 | scMoMaT overview. a** Graph illustration of running scMoMaT on an example dataset (4 batches and 3 modalities). Given data matrices in a mosaic layout to integrate, scMoMaT outputs cell representations and feature representations of multiple modalities, cell clusters and top-scoring bio-markers of every input modality. The top-scoring markers are used to annotate cell types for the clusters on the learned cell embedding. **b** Graph illustration showing the factorization of matrix $\mathbf{X}_{ij}$ in scMoMaT.

"feature" entities. A feature comes from a modality, which can be gene, region, or protein. An entity is the meaning of the rows or columns of each data matrix. For example, "cells batch1", "cells batch2", "genes", "regions", "proteins" are entities. We denote single cell data matrix from the $i$th cell batch and $j$th modality as $\mathbf{X}_{ij}$, where the rows of $\mathbf{X}_{ij}$ correspond to cells, and columns correspond to features of the modality. Then matrix tri-factorization decomposes $\mathbf{X}_{ij}$ into a cell factor $\mathbf{C}_i$, a feature factor $\mathbf{C}_j$, and an association matrix $\mathbf{\Sigma}_{ij}$. We add bias terms, $\mathbf{b}_i$ and $\mathbf{b}_j$, and scaling parameter, $\alpha_{ij}$, into the formulation, and the objective function for one data matrix is:

$$\hat{\mathbf{C}}_i, \hat{\mathbf{\Sigma}}_{ij}, \hat{\mathbf{C}}_j, \hat{\mathbf{b}}_i, \hat{\mathbf{b}}_j, \hat{\alpha}_{ij} = \arg\min_{\mathbf{C}_i, \mathbf{C}_j, \mathbf{\Sigma}_{ij}, \mathbf{b}_i, \mathbf{b}_j, \alpha_{ij}} \parallel \mathbf{X}_{ij} - \alpha_{ij}\mathbf{C}_i\mathbf{\Sigma}_{ij}\mathbf{C}_j^T - \mathbf{b}_i - \mathbf{b}_j^T \parallel_F^2 \quad (1)$$
$$s.t. \quad \mathbf{C}_x \cdot \mathbf{1} = \mathbf{1}, \mathbf{C}_x \geq 0, \mathbf{\Sigma}_{ij} \geq 0$$

$\mathbf{b}_i$ and $\mathbf{b}_j$ are 1 dimensional bias vectors for cell batch $i$ and modality $j$. The bias terms accommodate the data-matrix-specific information that cannot be encoded by the interaction between cell and feature factors. $\mathbf{b}_i$ encodes the data-matrix-specific variation among cells, and it has length equal to the number of cells. $\mathbf{b}_j$ encodes data-matrix-specific variation among features (which are genes if $\mathbf{X}_{ij}$ is a scRNA-seq count matrix), and it has length equal to the number of features. The scaling parameter $\alpha_{ij}$ is a scalar value, which accommodates the different scales of values in different data matrices. The row vectors of $\mathbf{C}_i$ and $\mathbf{C}_j$ respectively encode the latent factors of corresponding cells and features in the data matrix. In order for the latent factors to only capture the major biological variation within the data, the latent dimension $d$ (number of columns in $\mathbf{C}_i$ and $\mathbf{C}_j$) should be much smaller than the number of cells or features in the data matrix. We assume that each latent dimension (column vector) of $\mathbf{C}_i$ and $\mathbf{C}_j$ encodes a distinct biological factor of the dataset. Then the factor values of each cell or feature (each row vector) should encode the proportion of each biological factor contributing to the cell or feature identity, and they should be non-negative and sum up to 1 for each cell or feature. As a result, we constrain $\mathbf{C}_i$ and $\mathbf{C}_j$ with $\mathbf{C}_x \cdot \mathbf{1} = \mathbf{1}$, $\mathbf{C}_x \geq 0$. The association matrix $\mathbf{\Sigma}_{ij}$ encodes the interaction strength between cell and feature factors, where the value on the $r$th row and $c$th column correspond to the interaction strength between the $r$th dimension of cell factor and the $c$th dimension of gene factor. We constrain all values in $\mathbf{\Sigma}_{ij}$ to be non-negative. A graphical illustration of the factorization is shown in Fig. 1b.

When integrating multiple data matrices, we construct a loss function with multiple tri-factorization terms (Eq. (1)), where each input data matrix corresponds to a tri-factorization term. We use the scenario where the data matrices are from multiple batches and three modalities: gene, chromatin region and protein as an example (Fig. 1a). Denote the gene expression matrices as $\{\mathbf{G}_i\}_{i \in S_g}$, the chromatin accessibility matrices by $\{\mathbf{R}_j\}_{j \in S_r}$, and the protein abundance matrices by $\{\mathbf{P}_k\}_{k \in S_p}$. $S_g$, $S_r$ and $S_p$ are the sets of batch indices where gene expression, chromatin accessibility, and protein abundance matrices are available, respectively. The optimization problem of scMoMaT is:

$$\arg\min_{\mathbf{C}_x, \mathbf{\Sigma}} L$$
$$s.t. \quad \mathbf{C}_x \cdot \mathbf{1} = \mathbf{1}, \mathbf{C}_x > 0, \mathbf{\Sigma} \geq 0 \quad (2)$$

And

$$L = \sum_{i \in S_g} \parallel \mathbf{G}_i - \alpha_{ig}\mathbf{C}_i(\mathbf{\Sigma} + \mathbf{\Sigma}_{ig})\mathbf{C}_g^T - \mathbf{b}_{1i} - \mathbf{b}_{gi}^T \parallel_F^2 + \sum_{j \in S_r} \parallel \mathbf{R}_j - \alpha_{jr}\mathbf{C}_j(\mathbf{\Sigma} + \mathbf{\Sigma}_{jr})\mathbf{C}_r^T - \mathbf{b}_{2j} - \mathbf{b}_{rj}^T \parallel_F^2$$
$$+ \sum_{k \in S_p} \parallel \mathbf{P}_k - \alpha_{kp}\mathbf{C}_k(\mathbf{\Sigma} + \mathbf{\Sigma}_{kp})\mathbf{C}_p^T - \mathbf{b}_{3k} - \mathbf{b}_{pk}^T \parallel_F^2 + \lambda \left( \sum_{i \in S_g} \parallel \mathbf{\Sigma}_{ig} \parallel_F^2 + \sum_{j \in S_r} \parallel \mathbf{\Sigma}_{jr} \parallel_F^2 + \sum_{k \in S_p} \parallel \mathbf{\Sigma}_{kp} \parallel_F^2 \right)$$
$$(3)$$

where $\mathbf{C}_i$s are the factors for cell batches that have gene expression matrices, $\mathbf{C}_j$s are the factors for cell batches that have chromatin accessibility matrices, and $\mathbf{C}_k$s are the factors for cell batches that have protein abundance matrices. $\mathbf{C}_g$, $\mathbf{C}_r$ and $\mathbf{C}_p$ are the factors for genes, regions, and proteins. The factors of the same cell batch or feature modality are shared across the tri-factorization terms. $\mathbf{\Sigma}$ is the shared association matrix across all data matrices, and $\mathbf{\Sigma}_{ig}$, $\mathbf{\Sigma}_{jr}$, $\mathbf{\Sigma}_{kp}$ are data matrix-specific association matrices. $\mathbf{b}_{xx}$s are the cell or feature specific bias vectors for each data matrix. $\alpha_{ig}$, $\alpha_{jr}$, $\alpha_{kp}$ are data matrix-specific scaling parameters. $\lambda$ is the weight that regularize how much data matrix-specific association matrix should vary.

Each matrix encodes the relationship between the corresponding cell batch and feature entities, and Eq. (3) combines multiple data matrices which connect entities in these matrices through either direct or indirect relationships. For example, in Supplementary Fig. 1a, the protein and gene entities are connected through cell batch 1 using $\mathbf{P}_1$ and $\mathbf{G}_1$, similarly cell batch 1 and cell batch 2 are connected through gene entities using $\mathbf{G}_1$ and $\mathbf{G}_2$. However, there are cases where not all entities are connected with the existing data matrices. In Supplementary Fig. 1b, region entity cannot be connected to gene entity. In this case, we add pseudo-count matrices to connect all entities. In the specific scenario shown in Supplementary Fig. 1b, we calculate the pseudo-scRNA-seq matrix (or referred to as gene activity scores in some literature) from scATAC-seq matrix, with similar procedure to that used in Seurat and LIGER (Methods). We also describe how to calculate pseudo-protein-count matrices from scATAC-seq or scRNA-seq data matrices (Supplementary Fig. 1c, d, "Methods"). Such pseudo-count matrices are also required by existing mosaic integration methods[15–17], with some having stronger requirements than others. The pseudo-count matrices can also be used even when not theoretically required to strengthen cross-modality information and help with integration.

After the factors are learned by minimizing the objective function, we include an additional post-processing step on the learned cell factors (Methods). Similar post-processing steps have been used in existing integration methods that use matrix factorizations[5,16,19]. The post-processing step constructs a neighborhood graph of all cells, which can be visualized using UMAP and clustered using Leiden cluster algorithm[20]. After obtaining the cluster result of the cells, we retrain the model to learn the feature factors and association matrices. *Feature scoring matrices*, which represent the importance of a feature for a cluster, can then be obtained from the retrained feature factors and association matrices ("Methods"). These matrices have each latent dimension corresponding to one specific cell cluster, and can be used to extract the cluster-specific top-scoring features (bio-markers) across modalities that jointly define cell type identities.

### Testing scMoMaT on simulated datasets

First, we used simulated datasets to test scMoMaT, which allow us to generate different integration scenarios and quantitatively evaluate the integration method. The simulator that we used was similar to the simulator described in scDART[9], except that continuous cell populations were generated in scDART[9], whereas clusters of cells were generated in our tests. The simulation procedure can generate paired scRNA-seq, scATAC-seq, and protein abundance data (all modalities are profiled within each cell) from any number of batches ("Methods").

We generated 6 batches of paired scRNA-seq, scATAC-seq and protein abundance data, which results in 18 data matrices in total. To account for the randomness in the simulation, we repeat the simulation 8 times with 8 different random seeds and report summary results on the 8 datasets (Methods). For each dataset, there are 16 cell types shared across batches. We randomly selected 4 (out of 16) cell types for each data batch and removed these 4 cell types from the batch such that the batches have unequal cell type compositions (See Supplementary Table 1 for the numbers of cells and features for each batch in each dataset).

From these 18 data matrices, we created three different integration scenarios, as shown in Fig. 2a, b and Supplementary Fig. 3b. In

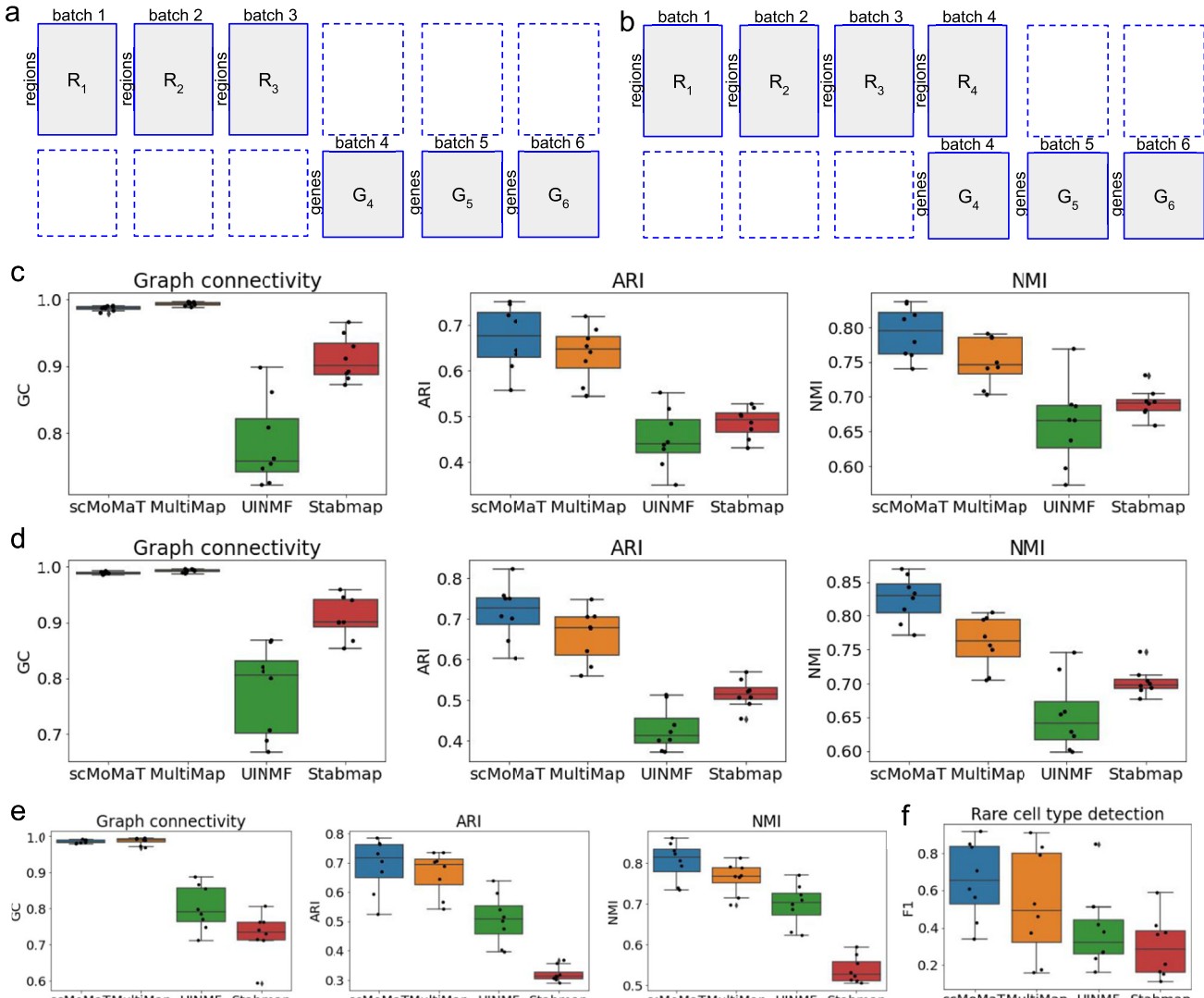

**Fig. 2 | Results on simulated datasets. a** The layout of data matrices under the first integration scenario. **b** The layout of data matrices under the second integration scenario. **c** GC, ARI, and NMI scores of scMoMaT and baseline methods under the first integration scenario. **d** GC, ARI, and NMI scores of scMoMaT and baseline methods under the second integration scenario. **e** GC, ARI, and NMI scores of scMoMaT and baseline methods under datasets with imbalanced batch sizes. **f** The

F1-scores showing the rare cell type detection accuracy of scMoMaT and baseline methods. In the boxplots above, the center lines show the median data value, and the box limits show the lower and upper quartiles (25% and 75%, respectively). The length of the whiskers is within 1.5× interquartile range. Outliers beyond the whiskers are plotted as points. $n = 8$ independent samples are included in each box. Source data for **c**, **d**, and **e** are provided in Source Data file.

Fig. 2a, no simultaneously profiled cell batch exists. We selected only the scATAC-seq matrices from batches 1, 2, and 3, and selected only scRNA-seq matrices from batches 4, 5, and 6 (totally 6 selected data matrices). In Fig. 2b, there exists one batch of cells simultaneously profiled with scATAC-seq and scRNA-seq (batch 4). In Supplementary Fig. 3b, we used three modalities of data. We test scMoMaT with different cases which may pose challenges for integration methods: (1) Unequal cell type compositions across batches; (2) Imbalanced sizes of data batches where the number of cells in different batches can be very different; (3) Rare cell types. Details on how to create these cases are in "Methods".

We compared the performance of scMoMaT with two recently published methods which can work with these integration scenarios: MultiMap[15], UINMF[16] and another mosaic integration method StabMap[17]. We ran scMoMaT, UINMF, MultiMap and StabMap under the first integration scenario (Fig. 2a). Firstly, we filled in the missing scRNA-seq matrices of the first three batches with pseudo-count matrices ("Methods"). Then, we ran scMoMaT and set its latent

dimension $d = 20$, number of neighbors $k = 30$, and radius parameter $r = 0.7$ for all runs. Details and parameter settings of MultiMap and UINMF are included in "Methods".

We quantitatively measured the overall performance of three methods with three scores: k-nearest neighbor graph connectivity (kNN-GC or GC), normalized mutual information (NMI), and adjusted Rand index (ARI) (Methods). These metrics were used in ref. [21] to benchmark various integration methods, where GC measures batch effect removal per cell identity label, and NMI and ARI measure conservation of biological variation during integration in terms of cell identity labels.

We summarized the performance of each method on 8 datasets using boxplots (Fig. 2c). The results show that scMoMaT performs comparably with MultiMap in matching cell batches (similar GC score), and perform consistently better in separating cell types (higher ARI and NMI scores). We visualized the latent embedding of scMoMaT and baseline methods on one of the 8 datasets using UMAP (Supplementary Fig. 2), and the visualization shows that with

scMoMaT the cell types are better separated, and the locations of the same cell type in the UMAP space are more consistent across batches. Taking cluster 16 which is missing in batches 2 and 3 as an example: in the results of MultiMap, cluster 16 is at consistent locations in batches 1, 4, 5, 6, but in batches 2 and 3, some cells from other clusters are placed at this location (circled in red). In the results of UINMF, cluster 16 in batch 1 is located in a different area from that in batches 4, 5, 6 (circled in red).

We then measured the performance of all four methods under the second integration scenario (as shown in Fig. 2b). We filled in the missing scRNA-seq matrices for the first three batches with pseudo-count matrices (the same as the first test scenario), and ran methods with hyper-parameter settings the same as the first test scenario. To make MultiMap applicable to the dataset, we concatenated the scATAC-seq and scRNA-seq in cell batch 4 into a single data matrix and reduced the dimensionality of that batch by running PCA on the concatenated matrix. Boxplots of GC, ARI and NMI scores of each method on 8 datasets are shown in Fig. 2d. The boxplots again show a better overall performance of scMoMaT compared to the two baseline methods.

Under the second scenario (Fig. 2b), we also test the ability of the integration methods in dealing with imbalanced sizes of cell batches and detecting rare cell types. When the cell batches have very different numbers of cells, scMoMaT is still the best method in terms of GC, ARI and NMI, and StabMap is the method whose performance is affected most by the imbalanced data size (Fig. 2e). We use F1 score (Methods) to measure the accuracy of rare cell type detection and scMoMaT has overall the best F1 score (Fig. 2f).

We then test the bio-marker detection accuracy of scMoMaT since the simulated data provides ground truth marker genes for each cluster (Methods). For baseline integration methods which does not output bio-markers, the marker genes are detected with an additional step of differential expression (DE) detection. We compare the accuracy of marker genes detected by scMoMaT with that from a baseline pipeline where we first use UINMF to learn the cell clusters in the integrated space and then use Wilcoxon rank-sum test to find DE genes between each cluster and the rest of the cells (Methods). The Wilcoxon rank-sum test is used to detect DE genes as it has been reported as one of the best DE detection methods[22,23]. For each cluster of cells, we then get three rankings of all the genes which represent how likely a gene is a marker gene: (1) The ground truth ranking; (2) The scMoMaT ranking obtained from the gene score; (2) The baseline ranking obtained by running UINMF and Wilcoxon test. We measured the accuracy using Kendall rank correlation coefficient between the ground truth and each of the other two rankings. The scores are summarized in the barplot in Supplementary Fig. 3a, where scMoMaT shows consistently better performance in marker detection compared to baseline method.

We further tested the performance of scMoMaT and baseline methods using all three modalities (Supplementary Fig. 3b), including chromatin accessibility, gene expression, and protein abundance. MultiMap and UINMF require that there exists a modality that is available in all batches. So we generated pseudo-protein count for batches 1, 2, 5, and 6 (Supplementary Fig. 3c, "Methods"). scMoMaT does not require pseudo-protein count matrices to be provided in this scenario, and we show the performance of scMoMaT with and without the pseudo-protein count matrices. Supplementary Fig. 3d shows the performance of scMoMaT with two modes (with and without pseudo-protein-counts) and baseline methods, where we can observe that scMoMaT consistently performs better than the baseline methods, and the inclusion of pseudo-protein count matrices further improves its performance.

## scMoMaT performs mosaic integration on human PBMC dataset and annotates sub-cell-types

We applied scMoMaT to a human PBMC dataset which includes 4 batches of cells[3]. The first 2 batches of cells are measured with gene expression and protein abundance simultaneously using CITE-seq[24,25] (batch 1 has 5023 cells, and batch 2 has 3666 cells); The last 2 batches of cells are measured with protein abundance and chromatin accessibility simultaneously using ASAP-seq[3] (batch 3 includes 3517 cells, and batch 4 includes 4849 cells). In total, there are 8 data matrices (Fig. 3a). On this dataset, we demonstrate: (1) scMoMaT produces integration with higher quality in terms of preserving cell identity and mixing batches compared to baseline methods; (2) scMoMaT improves cell type annotation through integration; (3) The bio-markers learned from multiple modalities lead to cell type annotation with higher resolution.

First, we visualized the cell factors of all batches learned by scMoMaT using UMAP (Fig. 3b, c). In Fig. 3b, the cells are colored with the cell type labels obtained from the original data paper[3], where different cell types are overall separated. In Fig. 3c, cells are colored by batches and cells from different batches are mixed in the integrated data. We compared the performance of scMoMaT with MultiMap[15], UINMF[16], and StabMap[17] (Details and parameter settings are included in Methods).

We measured the performance using metrics including GC, ARI and NMI scores. The original data paper[3] provides cell type labels for cells of all four batches, and these labels were used as ground truth clustering labels. The results (Fig. 3d) show that scMoMaT performs better than the three baseline methods with all metrics. Indeed, visualizations of latent embedding from MultiMap, UINMF, and Stab-Map (Supplementary Figs. 4a,b,c) show that the cell types were not matched as well between different batches for these methods.

Although it is a standard practice to test how much integration methods preserve the cell identities that were annotated in the original paper of the dataset (as shown in Fig. 3d)[15,21], we take a step further and ask whether we can improve this cell identity annotation through integrating data from multiple modalities. Performing Leiden clustering on the cell representations learned by scMoMaT, we obtained the clusters shown in Fig. 3e (upper plot) and mapped the labels to the clusters. We set to compare the labels from the original paper by Mimitou et al. and those obtained from scMoMaT. We consider that good cell labels should be consistent with the cell-cell variation in every batch and every modality. In Supplementary Fig. 5 we visualize each input data matrix respectively with the cell labels from scMoMaT and Mimitou et al. Visually, although in most of the plots different cell types are separated in the UMAP space, there are areas where more than one cell types are mixed (e.g. the areas circled in red). To quantify which set of labels has better agreement with the cell–cell variation in individual data matrix before integration, we used a metric named k-nearest neighbor agreement (kNN agreement). For each cell in each input data matrix, this metric measures the percentage of cells that have the same label as the given cell in its $k$ nearest neighbors (Methods). Figure 3e (lower plot) shows the kNN agreement score of each set of labels averaged over all cells in all input data matrices, where the scMoMaT labels have improved over the original labels from Mimitou et al.

We then show that the feature factors learned by scMoMaT give rise to bio-markers from multiple modalities which can be used to annotate cell types at higher resolution. We ran Leiden clustering algorithm on the integrated latent space of cells and obtained seven clusters (Fig. 3f, upper plot). We then fed the cluster labels into scMoMaT for retraining to obtain feature scoring matrices, which show the importance score of features in each cluster. Therefore, for each cell cluster, we have three vectors: (1) a vector representing the importance of each gene for this cluster; (2) a vector representing the importance of each chromatin region for this cluster; and (3) a vector representing the importance of each protein for this cluster. After we included the motif deviation matrix from chromVAR[18] analysis, we were also able to obtain vectors representing the importance of each motif for every cluster. The top-scoring features (genes, chromatin regions, proteins,

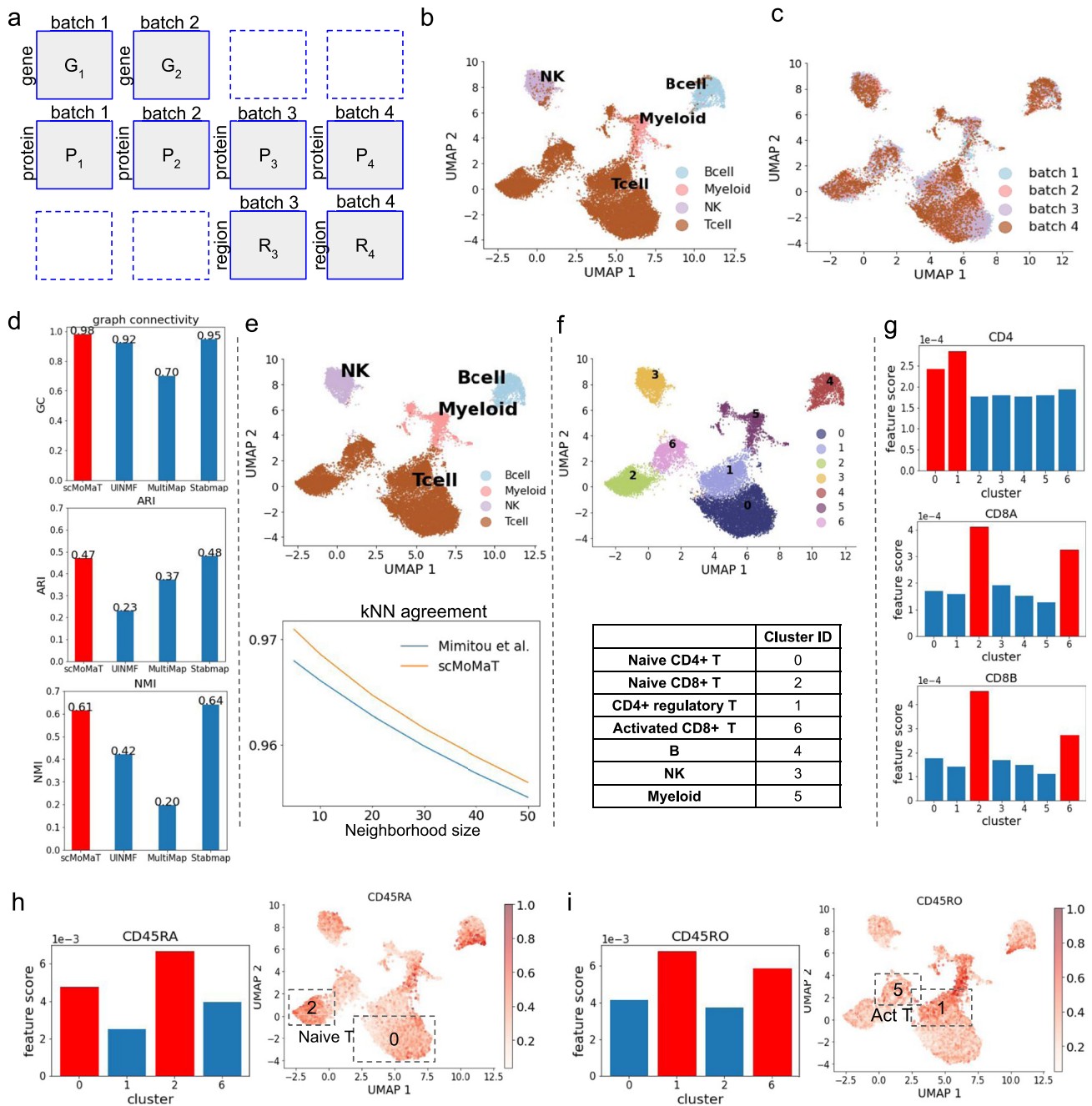

**Fig. 3 | Results on the human PBMC dataset. a** Layout of input data matrices in human PBMC dataset. The UMAP visualization of cell factors learned by scMoMaT, where cells are colored by (**b**) cell type labels from the original data paper (Mimito et al.) and (**c**) data batches. **d** The graph connectivity, ARI, and NMI scores of scMoMaT and baseline methods. The top-scoring method is colored red. **e** (Upper) The UMAP visualization of cell factors, where cells are colored by the label from scMoMaT. (Lower) The kNN agreement scores of scMoMaT labels and labels in original data paper (Mimito et al.) under different neighborhood sizes *k*. **f** (Upper) The UMAP visualization of cell factors; cells colored by Leiden clustering labels. (Lower) Cell type annotation for the Leiden clusters. **g** scores of marker genes *CD4*, *CD8A*, and *CD8B* in different clusters, where x-axis correspond to Leiden clusters. The top-scoring clusters are colored red. **h, i** The barplots show the scores of marker protein *CD45RA* and *CD45RO* in different clusters. The heatmaps show the abundance level of proteins *CD45RA* and *CD45RO*, where the top-scoring clusters are annotated in frames. Source data for b-i are provided in the Source Data file.

and motifs) in these vectors can be used as bio-markers for cell type annotation.

The complete annotation of these clusters is shown in Fig. 3f (lower table). We now discuss how the top-scoring features (with the highest importance score) from each modality lead to this annotation. First, we annotate clusters 3, 4 and 5 using top-scoring genes. The top-20 genes for cluster 3 include *GNLY*, *NKG7*, *KLRD1*, and *KLRF1*, which are the marker genes of Natural Killer (NK) cell[26,27]. The top-20 genes in

cluster 4 include *MS4A1*, *CD79A* and *CD37*, which are the marker genes of B cells[27,28]. The top-20 genes of cluster 6 includes *CTSS*[29], *SPI1*[30], and *CD63*[31], which are the marker genes of Myeloid cells (Supplementary Fig. 6a, marker genes with red frames). These annotations are further confirmed by extra marker genes for these cell types from CellMarker[32] (Supplementary Fig. 5a, known marker genes in blue frames). Furthermore, these annotations are consistent with the annotations from the original paper in visualization (Fig. 3b). These evidence together

show that the top-scoring genes learned by scMoMaT are highly consistent with known knowledge.

We have higher scores of CD3G, CD3E and CD3D (which are T-cell markers) in clusters 0, 1, 2, 6 than other clusters (Supplementary Fig. 6b), so we tentatively annotate these clusters as T cells. This is in agreement with the annotation in Fig. 3e. The feature factors learned by scMoMaT can be used to further identify T-cell subtypes in the integrated data. First, clusters 0, 1 have higher *CD4* scores, which shows that they correspond to CD4$^+$ T cells. Clusters 2, 6 have higher *CD8A* and *CD8B* scores, which shows that they correspond to CD8$^+$ T cells (Fig. 3g). The distributions of the expression value of *CD4*, *CD8A*, and *CD8B* also matches the importance scores of these genes (Supplementary Fig. 7a).

Within CD4$^+$ and CD8$^+$ T cells, top-scoring proteins can be used to further separate them into naive T cells and activated T cells. Naive T cells have high abundance of surface protein *CD45RA* and low abundance of surface protein *CD45RO*. Activated T cells, on the contrary, have high *CD45RO* and low *CD45RA*[33,34]. Using the scores of these proteins, we annotate clusters 0 and 2 to be naive T cells (lower *CD45RO* score and higher *CD45RA* score, Fig. 3h), and cluster 1, 6 to be activated T cell (higher *CD45RO* score and lower *CD45RA* score, Fig. 3i). The high importance scores of Naive T-cell marker genes (*CD27*, *TCF7*) in clusters 0 and 2 also confirms the annotation of Naive T cell from protein scores[34,35] (Supplementary Fig. 7b). Cluster 1 is further shown to correspond to CD4$^+$ regulatory T cell (Treg) using the importance scores of marker genes *Foxp3*, and *IL2RA*[35] (Supplementary Fig. 7c). Cluster 6 has high scores of cytotoxicity markers *GZMK* and *GZMB*[35] (Supplementary Fig. 7d), which further confirms the activated CD8$^+$ identity. Using the marker gene information in CellMarker[32], we found extra marker genes for the cell type annotated to clusters 0, 1, 2, 6 from the top-20 genes of these clusters (Supplementary Fig. 7e, with known marker genes in blue frames).

These discussions all together lead to the final annotations shown in Fig. 3f. The annotations are further confirmed by the known protein markers in the top-scoring proteins (Supplementary Fig. 8, with marker proteins in blue frames). Because scMoMaT also incorporates the motif deviation matrix learned by chromVAR[18] from the scATAC-seq data matrix, scMoMaT also outputs top-scoring motifs for each cluster. The known motif markers in the top-scoring motifs also confirm our cell type annotations (Supplementary Fig. 9, source of motif markers in Supplementary Data. 1).

## scMoMaT performs mosaic integration on mouse cortex data

We then applied scMoMaT on a mouse brain cortex dataset. We collected 5 batches of mouse brain cortex datasets from different publications. The first data batch has 10,309 cells where chromatin accessibility and gene expression were simultaneously measured using SNARE-Seq[2]. The second batch measures the gene expression of 40,166 cells using 10x v3 single-nucleus RNA-Sequencing technology (snRNA-seq) and the third batch measures the chromatin accessibility of 8718 cells using single-nucleus ATAC-Sequencing (snATAC-seq)[36]. The fourth batch measures the gene expression of 14,249 cells and is obtained from Allen Brain Atlas[37,38]. The fifth batch measures the chromatin accessibility of 3512 cells and is obtained from 10x Genomics website. In total, 6 data matrices are used as input to scMoMaT and they are organized as Fig. 4a.

First, to understand the variation structure between cells in each data matrix before integration, we visualized each data matrix separately using UMAP. The cells are colored using the cell type labels curated and re-organized from the original data paper (Methods, Supplementary Fig. 10). The visualizations show differences in the variation structures between different batches and modalities, which can be caused by various factors, including technical confounders such as read depth and noise level[39,40], or disparity of cell type composition between batches. Applying scMoMaT to these matrices leads to integrated cell representations in a shared latent space, and the top-scoring features

output from scMoMaT can be used as bio-markers for cell type annotation. Through the latent space representations of genes and regions learned by scMoMaT, we also demonstrate that bio-markers for the same cell type tend to have similar low-dimensional representations.

scMoMaT took as input the six data matrices and two additional pseudo-scRNA-seq matrices that were calculated from scATAC-seq matrices for data batches 3 and 5 (Methods). The learned cell representations are shown in Fig. 4b, c, where the clusters in Fig. 4b were obtained by running Leiden clustering[20] on the cell factors, and the cell types were annotated with bio-markers learned by scMoMaT. The cell type annotation process is described below.

We first collected known marker genes for the cell types included in these datasets from existing literature[36,41,42] (Supplementary Table 2). The scores of these marker genes learned through the retraining step were used to annotate the clusters in Fig. 4b: for the non-neuron cell types, *Mbp* and *Plp1* annotate cluster 9 as oligodendrocyte (Oligo), *Aldoc* and *Slc1a3* annotate cluster 8 as Astrocytes (Astro), *Csf1r* and *C1qb* annotate cluster 12 as Macrophage, *Matn4* and *Lhfpl3* annotate cluster 11 as oligodendrocytes (OPC) (Fig. 4d). For the neuronal cell types, L6 neuron marker gene *Sulf1* has high scores in clusters 0, 2, and 10 (Fig. 4e). Out of these three clusters, *Foxp2* was used to distinguish L6 corticothalamic neuron (L6 CT/b, cluster 2 and 10, with high *Foxp2*) from L6 intratelencephalic neuron[36] (L6 IT, cluster 0, with low *Foxp2*, Fig. 4e). The scores of *Rorb* annotate cluster 3 as L4/5 excitatory neuron[36], *Tshz2* annotate cluster 7 as near-projecting excitatory neurons (NP), and *Calb1* annotate cluster 1 as L2/3 excitatory neurons (Fig. 4e). We also found high scores of marker genes *Pvalb*, *Sst*, *Npy*, *Vip* in clusters 4 and 6, which shows that those two clusters corresponds to GABAergic inhibitory neurons (Fig. 4e). Cluster 4 has higher scores of *Pvalb* and *Sst* and cluster 6 has higher scores of *Npy* and *Vip*, which shows that these two clusters correspond to distinct sub-cell types in GABAergic inhibitory neurons[43] (Fig. 4e).

The top-20 scoring genes for each cluster are enriched with known marker genes for the annotated cell type, according to marker genes collected in Supplementary Table 2 and in CellMarker[32] (Supplementary Fig. 11, with known marker genes in blue frames). There are fewer marker genes found for neuronal cell subtypes partly because fewer marker genes are known for these cell types.

Including the motif deviation matrix (from chromVAR) allows scMoMaT to learn top-scoring motifs for each clusters (Fig. 5a). Out of the top-20 motifs, we see MA0062.2_Gabpa, MA0117.2_Mafb and MA0002.2_RUNX1 for Macrophage (cluster 12), MA0515.1_Sox6, MA0442.1_SOX10 and MA0514.1_Sox3 for oligodendrocyte (cluster 9), MA0463.1_Bcl6 and MA0518.1_Stat4 for L6 CT/b neuron (clusters 2 and 10), etc[44]. In particular, motifs MA0623.1_Neurog1 and MA0461.2_Atoh1 stand out in L6 CT/b, and *Neurog1* and *Atoh1* are reported to be important transcription factors in neurogenesis[45,46]. Overall, these motifs further support our cell type annotations.

Since scMoMaT jointly learns the region and gene factors along with the cell factors, we also visualize the region and gene factors (Fig. 5b, c). Figure 5b shows the gene factors where known marker genes for different cell types are marked with different colors. One can observe that the marker genes for GABAergic inhibitory neurons, oligodendrocyte, oligodendrocyte precursors, Macrophage, Astrocytes, and Glutamatergic neurons (including L2/3, L4/5, L6 IT, L6 CT/b, NP) are clearly separated into different areas of the UMAP space. For the region factors, we map a region to a gene if the region is located within the 2000 base-pair upstream or the gene body of the gene on the genome, and we plot the genes as the average of the regions associated with the corresponding gene (genes are represented by colored dots in Fig. 5c). Figure 5c shows that the chromatin regions that correspond to marker genes of oligodendrocyte & oligodendrocyte precursors, Macrophage, and Glutamatergic neurons are also separated into distinct areas based on the region factors. Both the gene and the region factors

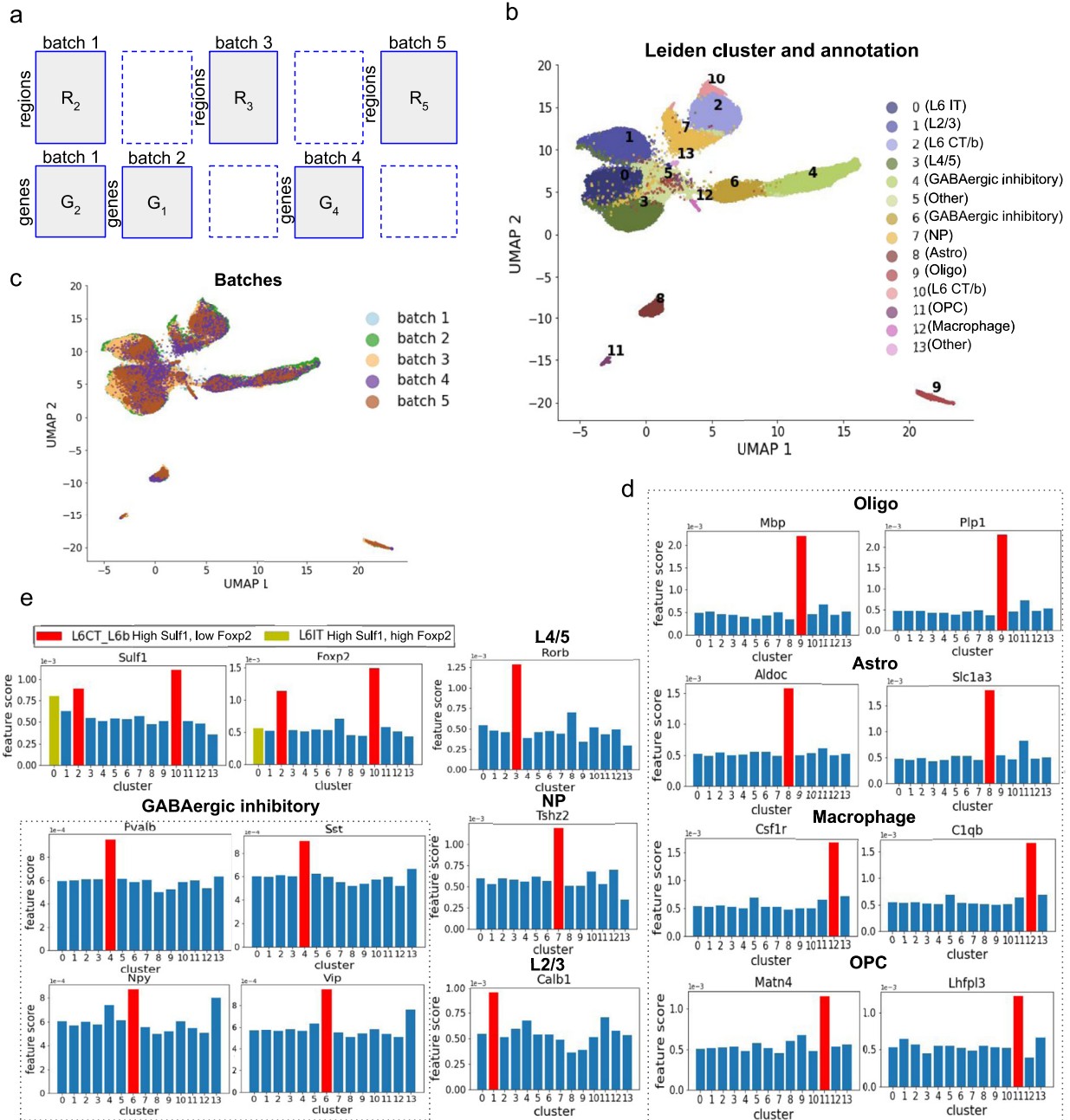

**Fig. 4 | Results on mouse brain cortex dataset. a** Layout of input data matrices in mouse brain cortex dataset. The UMAP visualization of cell factors learned by scMoMaT, where cells are colored by **b** Leiden clusters (with scMoMaT-annotated cell types) and **c** cell batches. **d** The scores of marker genes for non-neuronal cell types, where x-axis correspond to Leiden clusters. **e** The scores of neuronal cell type marker genes in different clusters. The top-scoring clusters are colored red. Source data for **b**–**e** are provided in the Source Data file.

show that genes and regions do not form distinct clusters (which is expected because genes like house keeping genes do not belong to a specific gene module), but marker genes of different cell types are separated in the gene and region factor space learned by scMoMaT.

## scMoMaT integrates batches with no shared modalities

It is a very challenging integration scenario if the batches do not share any modality (also called diagonal integration). The most common example of such a scenario is the integration of a scATAC-seq matrix and a scRNA-seq matrix obtained from different batches. To integrate such datasets, additional assumptions or information often need to be provided. Some methods assume that the latent distributions of cells

are similar between batches, which fails to accommodate the cases where the data batches have unequal cell type compositions. Other methods transform the scATAC-seq matrix into a pseudo-scRNA-seq matrix using the cross-modalities relationship, and integrate the scRNA-seq matrix and pseudo-scRNA-seq matrix. Using the pseudo-scRNA-seq instead of the scATAC-seq matrix, these methods may suffer from the errors introduced during the process of calculating the pseudo-scRNA-seq matrix and do not fully utilize the epigenomic information in the scATAC-seq matrix. scMoMaT, on the other hand, keeps both the original scATAC-seq matrix and the pseudo-scRNA-seq matrix in order to better exploit the scATAC-seq information. Also, we binarized the pseudo-scRNA-seq matrices as a denoising step (Methods).

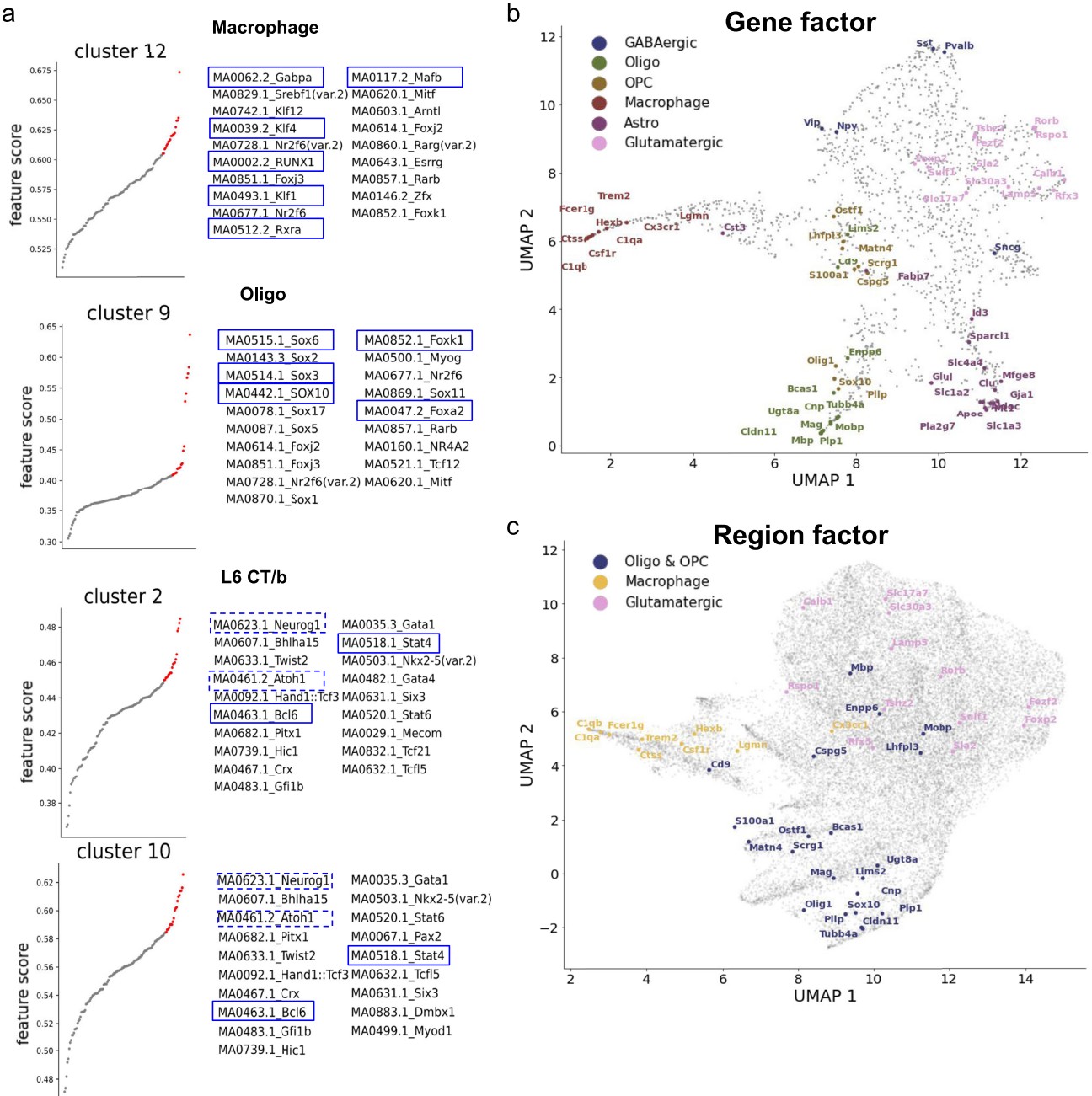

**Fig. 5 | Additional results on mouse brain cortex dataset. a** The top-20 scoring motifs in cluster 12 (Macrophage), 9 (Oligo), and 2 (L6 CT/b). Motifs with TFs reported for a specific cell type are highlighted in blue frames. The UMAP visualization of **b** gene factors and **c** region factors; known marker genes of different cell types are annotated with corresponding colors. Source data for a are provided in the Source Data file.

We applied scMoMaT to a healthy human bone marrow mononuclear cells (BMMC) dataset[47]. The dataset includes two batches of cells, where the first batch has 16,510 cells sequenced with scATAC-seq and the second batch has 12,601 cells sequenced with scRNA-seq (matrix relationships follows Fig. 6a). scMoMaT takes as input both matrices, and generates a pseudo-scRNA-seq matrix for the second batch using its scATAC-seq data matrix (Methods). We visualize the cell factors learned from scMoMaT (Fig. 6b, c) using UMAP, and color the cells using the literature-derived labels (Fig. 6b) and data batches (Fig. 6c). In the visualizations, cell batches are well integrated in the latent space, while cell identities in each data batch are also preserved.

We also ran UINMF, MultiMap, LIGER[5], Seurat[10] and StabMap[17] on the dataset (results visualized in Supplementary Fig. 12). We

quantitatively measured the overall performance of the methods using GC, NMI, and ARI scores ("Methods", Fig. 6d). scMoMaT has the highest GC score, which shows that scMoMaT better matches the same cell type between batches. scMoMaT and UINMF have similar ARI and NMI scores. The ARI and NMI scores of Seurat are slightly higher than both scMoMaT and UINMF, while the scores of LIGER and StabMap are worse than scMoMaT and UINMF. Overall, scMoMaT and Seurat are two top performers on this dataset with comparable results. MultiMap, on the other hand, mixes cells from different cell types in the latent space. It may be due to the fact that the cell types in BMMC dataset are closely located in the original dataset as they follow the trajectories of the hematopoiesis process, and MultiMap fails to distinguish the closely located cell types (Supplementary Fig. 12b).

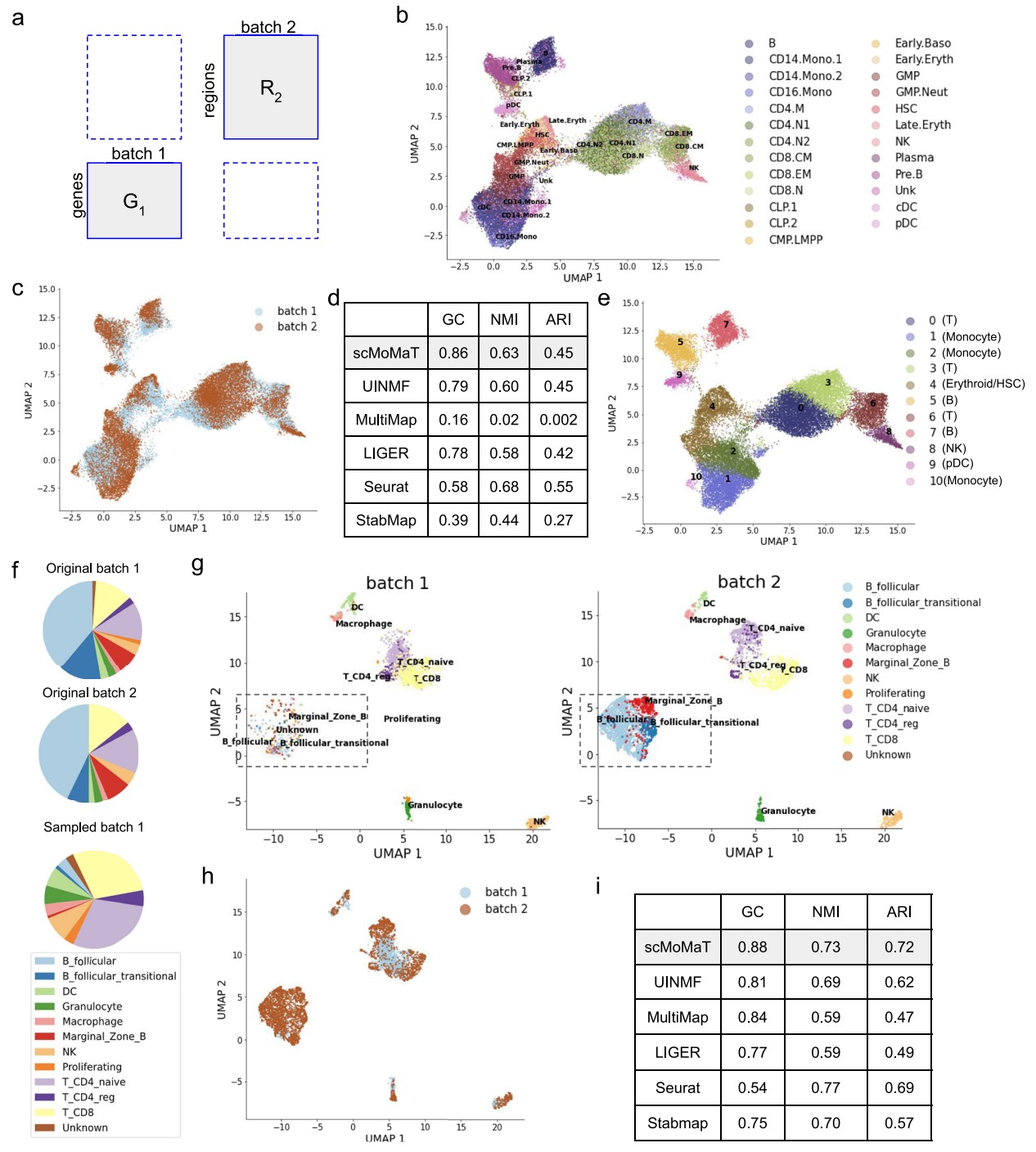

**Fig. 6 | Results on the human bone marrow and mouse spleen dataset.**
**a** Relationship between data matrices in the two datasets. The UMAP visualization of cell factors, where cells are colored by (**b**) ground truth cell type, and (**c**) data batches. **d** The GC, ARI, and NMI scores of scMoMaT and baseline methods. **e** The UMAP visualization of cell factors, where cells are colored by Leiden clusters (with scMoMaT-annotated cell types). **f** The cell type composition of each batch in original and subsampled mouse spleen dataset. **g** The UMAP visualization of cell factors learned from subsampled dataset; Batches 1 and 2 are visualized separately. The disproportionate B follicular cells are annotated in frames. **h** The UMAP visualization of cell factors learned from subsampled dataset, where cells are colored by data batches. **i** The graph connectivity (GC), ARI, and NMI scores of scMoMaT and baseline methods on the subsampled dataset. Source data for **b**, **c**, **e**, **f**, **g**, and h are provided in the Source Data file.

After clustering the cells, scMoMaT learned the importance scores of genes and regions in each cluster. We annotated cell types according to the top-scoring genes and regions. The cluster result and cell type annotations are shown in Fig. 6e. Since we also input the motif deviation matrix obtained by chromVAR, scMoMaT learns top-scoring motifs along with genes and regions. Multiple known marker genes

and relevant motifs are shown to have high scores for their corresponding cell types. In cluster 8 (natural killer (NK) cells), scMoMaT found a high score of marker gene *GNLY*[26,27], which matches the gene expression pattern in the dataset (Supplementary Fig. 13a). Similarly, scMoMaT found T-cell marker gene *CD3D*[35] in clusters 0, 3, and 6 (Supplementary Fig. 13a), B-cell marker gene *CD79A*[28] in clusters 5 and

7 (Supplementary Fig. 13a), Monocyte marker gene *S100A9*[48] in clusters 1, 2, and 10 (Supplementary Fig. 13a), and Plasmacytoid Dendritic Cell (pDC) marker gene *PTPRS*[49] in cluster 9 Supplementary Fig. 13a). The top-20 genes of clusters 0, 3, and 6 reveal even more T-cell-related marker genes including *BCL11B*, *IL7R*, *LEF1*, etc[32] (top-20 genes in Supplementary Fig. 13b, with known marker genes in blue frames).

Meanwhile, the top motifs for each cluster also confirmed the cell type annotations. scMoMaT found high scores of motif *MA0102.3_CEBPA*, *MA0837.1_CEBPE*, and *MA0466.2_CEBPB* in Monocytes. Their corresponding transcription factors *CEBPA*, *CEBPE*, and *CEBPB* are known to be monocyte-differentiation regulators[47,50,51] (Supplementary Fig. 14a). In addition, motifs *MA0800.1_EOMES*, *MA0802.1_TBR1*, and *MA0690.1_TBX21* have high scores in cluster 8 (NK). *EOMES* regulates the maturation of NK cells[52], whereas *TBX21* (also known as T-bet, belonging to T-box subfamily *TBR1*) is also known to orchestrate the development and effector functions in NK cells[53] (Supplementary Fig. 14b).

Finally, the cell type annotations obtained with the learned bio-markers are overall consistent with the original cell type annotations (Fig. 6b, e), which verifies that the bio-markers we learned are meaningful.

### scMoMaT integrates batches with unequal cell type compositions

In this section, we test how well scMoMaT performs when the cell type compositions are unequal between batches. We used a mouse spleen dataset[15,54] that includes two batches of cells, where the first batch has 4382 cells sequenced with scRNA-seq, and the second batch has 3166 cells sequenced with scATAC-seq (matrix relationships follows Fig. 6a). The dataset mainly consists of T cells (1190 cells in Batch 1, 990 cells in Batch 2), B cells (2621 cells in Batch 1, 1835 cells in Batch 2), and some other cell types that reside in mouse spleen. The original two data batches have similar cell type compositions (Fig. 6f). We created data batches with disproportionate cell types by subsampling the most populated cell type, B cells (including B_follicular, B_follicular_transitional and Marginal_zone_B), in Batch 1 such that only 100 B cells were left. The subsampling step changed the proportion of B cells from 59.8% to 5.4%, which drastically changed the cell type composition of Batch 1 (Fig. 6f).

We applied scMoMaT, UINMF, MultiMap, LIGER, Seurat and StabMap to this dataset. The visualization shows that scMoMaT can correctly match cell types in two data batches regardless of the disproportionate cell type compositions between two batches, especially B cells which barely exist in the first batch (Fig. 6g,h). The cell factors of two batches are separately plotted in Fig. 6g for the two batches, where B cells lie within the box. UINMF, StabMap, and MultiMap also perform reasonably well in terms of integrating the two batches (Supplementary Figs. 15 and 16), but the cell types are not clearly separated in MultiMap (Supplementary Figs. 15, 16, 6i). Seurat has a slightly higher NMI score than scMoMaT, but its GC score is much lower than scMoMaT. Overall, scMoMaT has the best overall performance among these methods on this dataset (Fig. 6i).

### Effects of hyper-parameters settings in scMoMaT
The hyper-parameters in scMoMaT include the latent dimension $d$ for cell and feature factors, the regularization weight $\lambda$ in the loss function, the number of neighbors $k$ and radius parameter $r$ in the post-processing step.

We tested how scMoMaT performance is affected by different hyper-parameter settings using the simulated datasets illustrated in Fig. 2b. We measured the performance of scMoMaT using three metrics, including NMI, ARI, and GC. We first tested the two parameters used in the training stage including $d$ and $\lambda$, where $d = \{10, 20, 30, 50\}$ and $\lambda = \{10^{-4}, 10^{-3}, 10^{-2}\}$. Supplementary Fig. 17a shows that all three metrics are not sensitive to the change of $\lambda$. The metrics are also

robust with the change in $d$ except for the case where $d = 10$, as 10 is too small a number for the latent dimensions

We then varied the two hyper-parameters used in post-processing stage including $k$ and $r$, where $k = \{15, 30, 50\}$ and $r = \{0.7, 0.9, 1\}$. In Supplementary Fig. 17b, we observe that the metrics are not sensitive to the change of $k$. We also observe that ARI and NMI decrease under larger $r$, although only by a small amount. This is because these data-sets have unmatched cell types across data batches, and a smaller $r$ can better accommodate for cell type mismatch. When the datasets are expected to have equal cell types, we suggest setting $r = 1$ to skip the pruning step (Methods).

We then conducted hyper-parameter testing on the human PBMC dataset[3] using the same values for $\lambda$, $r$ and $k$ as used on the simulated datasets, and extended the values of $d$ to $d = \{10, 20, 30, 50, 80\}$ since real datasets tend to require more latent dimensions than simulated data (Supplementary Fig. 18). From these results, we see a pattern that is consistent with observations from Supplementary Fig. 17: scMoMaT is overall very robust to the changes in $\lambda$, $k$ and $r$. In terms of $d$, in both Supplementary Figs. 17 and 18, the NMI and ARI scores stabilize once $d$ reaches a certain threshold, which is a number of latent dimensions that is large enough for the given dataset. This threshold is smaller in simulated data than that in real data, which is expected.

We also provide guidance on hyper-parameter selection, as well as the settings of hyper-parameters in all test results that are presented in this manuscript ("Methods").

## Discussion
In this study, we introduced scMoMaT, a single cell data integration method that works on mosaic integration scenario. We applied scMoMaT on different mosaic integration tasks. The results validated the broad applicability of scMoMaT under various types of data integration scenarios. We showed that scMoMaT not only has superior performance compared to existing methods in terms of metrics used to evaluate integration methods, but also learns cluster-specific bio-markers from every input modality that can be used to annotate cell types in the integrated cell space with high confidence. The new annotations can improve the annotations provided in the original papers that publish the datasets, as the clustering and annotations from scMoMaT comprehensively consider information from multiple modalities. Furthermore, we also showed that scMoMaT is able to integrate batches that have disproportionate cell type compositions. With the increasing availability of single cell multi-omics datasets, we expect that scMoMaT will be widely applied to various data integration tasks.

Compared to data integration methods that only learn cell representations in the integrated space, scMoMaT also learns feature representations (e.g., gene representations). In the future, considering feature representations in the data integration framework can help with learning cross-modality relationships from single cell multi-omics data.

## Methods
### Training procedure of scMoMaT
We minimize the loss function of scMoMaT (Eq. (2) and (3)) using mini-batch stochastic gradient descent. Within each iteration, we pick one parameter matrix from cell and feature factors (the $\mathbf{C}_x$ matrices), shared and data matrix-specific association matrices ($\{\Sigma, \Sigma_{xx}\}$), bias matrices ($\mathbf{b}_{xx}$), and scaling parameter $\alpha_{xx}$ and fix the other parameter matrices. Then, we update a mini-batch of the selected parameter matrix using gradient descent. Each mini-batch is constructed by subsampling 10% of cells and features in each data matrix. Then, we loop through all parameter matrices and update them using gradient descent in order.

In order to enforce the simplex constraint on the factor matrices, we transform the original factor matrices using a softmax function

before using it to calculate the reconstruction loss and use the softmax-transformed factor matrices as the output factor matrices of the model. We enforce the non-negativity constraint on the shared association matrix $\Sigma$ by changing all its negative values to zero every time that it is updated.

In each iteration, we update the bias terms and scaling parameters using closed-form solutions by setting its gradient to 0. Taking the data matrix $G_i$ as an example, the closed-form solution of its cell bias term $b_{1i}$ follows:

$$b_{1i} = \frac{1}{n_{\text{feats}}} \sum_{n=1}^{n_{\text{feats}}} (G_i[:,n] - \alpha_{ig} C_i(\Sigma + \Sigma_{ig})C_g^T[:,n] - b_{gi}[n]) \quad (4)$$

where $n_{\text{feats}}$ is the total number features in $G_i$. Similarly, its feature bias term $b_{gi}$ follows:

$$b_{gi} = \frac{1}{n_{\text{cell}}} \sum_{n=1}^{n_{\text{cell}}} (G_i[n,:] - \alpha_{ig} C_i(\Sigma + \Sigma_{ig})C_g^T[n,:] - b_{1i}[n])^T \quad (5)$$

where $n_{\text{cell}}$ is the total number of cells in $G_i$. The scaling parameter can be calculated as

$$\alpha_{ig} = \frac{tr(G_i - b_{1i} - b_{gi}^T)^T (C_i(\Sigma + \Sigma_{ig})C_g^T)}{tr((C_i(\Sigma + \Sigma_{ig})C_g^T)^T (C_i(\Sigma + \Sigma_{ig})C_g^T))} \quad (6)$$

The complete pseudo-code of scMoMaT is included in Supplementary Note 1.

## Calculating pseudo-count matrices

scMoMaT uses the relationship between different feature entities to create pseudo-count matrices that can fill in the positions of missing modalities or batches during integration. For each feature in the target modality, its pseudo-count is calculated by combining the counts of the features in the original modality that only correlated with the target feature. We further explain the calculation step of pseudo-count matrices using three example modalities that are most commonly used: chromatin accessibility, gene expression, and protein abundance.

When calculating pseudo-scRNA-seq matrix from scATAC-seq matrix, scMoMaT constructs pseudo-scRNA-seq matrix from scATAC-seq matrix by summing up the region counts from all regions that lie within the 2000 base-pair upstream from the TSS of the gene and the regions that lie within the gene body on the genome. Different from the gene activity matrix that was used in Seurat and LIGER, scMoMaT further binarizes the pseudo-scRNA-seq instead of directly using it for integration. The use of additional binarization step is based on two reasons: (1) the relationship between region counts and gene counts is not linear. The activation of the promoting regions of a gene correlate with the activation of the transcription process of the gene, but there is not enough evidence showing that the gene expression level is positively correlated with the number of activated promoting regions. (2) binarized gene counts are shown to also have enough ability in distinguishing cell types[55].

When calculating pseudo-protein count matrix from scRNA-seq matrix, scMoMaT first connects each protein with its corresponding RNA molecule as there exists a one-one correspondence between the two modalities. Then for each protein, we use the gene expression count of the corresponding RNA molecule as the pseudo-protein count.

When calculating the pseudo-protein count matrix from scATAC-seq matrix, scMoMaT first calculates the pseudo-scRNA-seq from scATAC-seq following the first example above, then calculate the pseudo-protein count from pseudo-scRNA-seq count following the second example above.

## Post-processing procedure

After training the model, we calculate a pairwise distance matrix between cells from all batches using cell factor values. We then construct a neighborhood graph from the distance matrix by connecting each cell with both its within-batch nearest neighbors and its cross-batches nearest neighbors. Denoting the overall number of nearest neighbors for each cell by $k$ ($k = 30$ for most of the results shown), the number of nearest neighbors taken in each batch is proportional to the total number of cells in the batch. More specifically, the number of neighbors $k_i$ for batch $i$ can be calculated by $k_i = (N_i/N_{\text{total}}) \cdot k$, where $N_i$ is the number of cells in batch $i$, and $N_{\text{total}}$ is the total number of cells in all batches. We also offer an option to prune the connections in the neighborhood graph using a radius parameter $r$. The radius parameter is from 0 to 1, denoting the percentage of connections to be preserved between every two batches.

After obtaining the neighborhood graph, we then normalize the distances between cells in the graph. We first calculate the mean within-batch distance and mean cross-batches distances for each cell using the distance of the cells to its within-batch nearest neighbors and cross-batches nearest neighbors. Then we normalize the distances between the cell and its cross-batches nearest neighbors, which makes the mean within-batch distance and mean cross-batches distances for the cell to be the same. Considering cell $m$ and cell $n$ are nearest neighbor calculated from batch $i$ and batch $j$, the distance $d_{mn}$ between $m$ and $n$ can be normalized by $\hat{d}(mn) = (\bar{d}_{ii}/\bar{d}_{ij})d(mn)$, where $\hat{d}(mn)$ is the normalized distance between cell $m$ and cell $n$, $\bar{d}_{ii}$ is the mean within-batch distance of cell $m$ and its neighbors in batch $i$, $\bar{d}_{ij}$ is the mean cross-batches distance of cell $m$ and its neighbors in batch $j$. The normalized neighborhood graph can be used for visualization and clustering purposes. UMAP can take the neighborhood graph to visualize the cell-to-cell variation, and Leiden clustering algorithm is used to cluster the cells based on the neighborhood graph.

## Retraining procedure

After clustering the cells, we use the cluster label for the retraining of scMoMaT. We first construct binary cell factor matrices from the cluster label by making each column dimension of the cell factor matrices match one specific cell cluster, and by assigning 1 to the corresponding cluster dimension and 0 to the other dimensions for each cell. The retraining step is to learn feature factors and association matrices that are consistent with the binary cell factors.

We then fix the binary cell factor matrices and update the remaining parameters in scMoMaT to minimize the loss (Eq. (2)). The retrained feature factor matrices and association matrices can be used to build the feature scoring matrices that includes the marker score for each feature in each cell cluster. The top-scoring features in each cluster are considered to be the bio-markers of the cluster. Given the retrained feature factor matrix $C_{\text{feat}}$ (e.g. $C_g$, $C_r$, $C_p$) and shared association matrix $\Sigma$, the feature scoring matrix $M_{\text{feat}}$ can be calculated as $M_{\text{feat}} = C_{\text{feat}} \cdot \Sigma^T$, and each column of $M_{\text{feat}}$ are the marker scores of all features in the corresponding cell cluster.

During the retraining process, scMoMaT is flexible on the data matrices that are used for each data batch. One can incorporate additional data matrices that measure different data modalities of the existing data batches into the retraining process and learn the factor of the newly added data modalities through scMoMaT. In the testing result of mouse brain cortex dataset, PBMC dataset, BMMC dataset, we obtained the motif deviation matrices (cell by motif matrices, calculated from scATAC-seq matrix using chromVAR), and included the motif deviation matrices in the retraining process to learn the motif factor of the dataset.

## Hyper-parameter setting in scMoMaT

There are four hyper-parameters in scMoMaT: the latent dimension $d$ for cell and feature factors, the regularization weight $\lambda$ in the loss

function, the number of neighbors $k$ and radius parameter $r$ in the post-processing step.

The latent dimension $d$ corresponds to the number of latent biological factors that should be included in the dataset. It varies according to the complexity of the cell-cell variation in the dataset. The higher the complexity is, the larger $d$ is required. It does not correspond to the number of cell types and is usually larger than the number of cell types. In all our tests on real datasets, we select $d = 30$. In all our tests on simulated datasets, where the dataset structure is less complex, we set $d = 20$. The regularization weight $\lambda$ is selected to be 0.001 (default value) for all our tests.

The number of neighbors $k$ in the post-processing step should be based on the total number of cells (larger $k$ for larger dataset). We suggest users to set $k$ to be 30–50. The radius parameter $r$ prunes weak connections in the neighborhood graph. Smaller $r$ means more connections are removed. We suggest to apply the pruning step when there is a strong mismatch between cell type composition of different cell batches. We did not apply pruning for the real datasets; and on simulated datasets, we applied pruning and set $r = 0.7$ because of the mismatch of cell types between cell batches.

## Data simulation

We implemented a simulation procedure that can generate multiple batches of paired scRNA-seq, scATAC-seq, and protein abundance datasets which is generalized upon previous work SymSim[23]. In the same batch, cross-modality relationship between scATAC-seq and scRNA-seq data, and between scRNA-seq and protein abundance data are modeled in this simulator. The relationships between scATAC-seq and scRNA-seq data are considered through the kinetic model used to generate the scRNA-seq data. More details on this procedure are described in ref. [9]. The process of generating the protein modality from scRNA-seq modality is described in Supplementary Note 2. The numbers of genes, regions, proteins, and cells in each batch of each simulated dataset are shown in Supplementary Table 1.

On the simulated datasets, we created various challenging cases to test scMoMaT and the baseline methods: (1) Unequal cell type compositions across batches. We randomly selected 4 (out of 16) cell types for each data batch and removed these 4 cell types from the batch such that the batches have unequal cell type compositions. (2) Imbalanced sizes of data batches where the number of cells in different batches can be very different. We create the imbalanced dataset by subsampling the cells in the original simulated datasets for 3 batches. For cell batches 1, 4, and 6 in each dataset, we subsampled their cells such that each of these three batches has a number of cells that is a tenth of each of the remaining batches. (3) Rare cell types. In each batch of cells before removing the 4 cell types to create the unequal cell type composition scenario, there are 16 cell types and one of them is a rare cell type. The number of cells in the rare cell type is 15% of the number of cells of a normal cell type.

To evaluate the bio-marker detection accuracy of scMoMaT, we used the ground truth marker genes available from the simulation. The SymSim package allows users to output ground truth differentially expressed genes between two groups of cells based on mRNA counts without intrinsic and technical noise. The ground truth is in the form of a ranking of genes on how likely a gene is a DE gene. To obtain DE genes from the baseline method, we use the common pipeline in most of the current data integration methods: (1) We learned the cell latent factor from the dataset using UINMF. (2) We clustered the cells by running Leiden cluster algorithm on the cell factor. (3) We conducted two-sided Wilcoxon rank-sum test to detect cluster-specific bio-markers. When running Leiden clustering algorithm, we select the resolution parameter that gives the highest NMI score.

## Preprocessing of datasets

When running scMoMaT on real datasets, we filter genes in the scRNA-seq matrices by selecting highly variable genes. Then we quantile normalize the scRNA-seq matrices. We quantile normalize and log-transform the protein abundance matrices[56]. No protein filtering step is conducted as there is a small number of proteins measured. The scATAC-seq is filtered by selecting the regions that lie within the 2000 base-pair upstream activation region and the gene body of all genes kept in the scRNA-seq count matrices. When dealing with multiple scATAC-seq matrices with different region features, we remap the fragment file of other scATAC-seq matrices using the peaks from one scATAC-seq matrix that we select, which was also used in Cobolt[14] and Signac[57]. This allows the scATAC-seq matrices to have the same region features.

With simulated datasets, we did not filter the genes or regions for all integration methods. The pseudo-scRNA-seq matrices are calculated by multiplying the "region by gene" association matrix that is provided by the simulator with the scATAC-seq matrices. The pseudo-protein abundance matrices are calculated by multiplying the "protein by gene" association matrix with scRNA-seq matrices or by multiplying the "protein by region" association matrix with scATAC-seq matrices. We quantile normalize the scRNA-seq matrix, quantile normalize and log-transform the protein abundance matrix, and binarize the scATAC-seq matrix when inputting these data to scMoMaT. When running UINMF and MultiMAP on the simulated data, we followed the online tutorial of these methods (see Section "Running baseline methods").

Some details in the preprocessing procedures can vary for each real dataset, which are described as follows.

**Human PBMC dataset.** For the human PBMC dataset, we selected top 7000 highly variable genes using scanpy for each scRNA-seq matrix separately. We do not remap the scATAC-seq matrix as the dataset comes with the same region features in the two scATAC-seq matrices. After gene and region filtering, we obtained overlapping 4768 genes, 17,442 regions and 216 proteins.

**Mouse brain cortex dataset.** Since the scATAC-seq matrices have different sets of region features, we first remapped the scATAC-seq matrices in the first and the fifth batches to the regions in the third batch. We then selected the top 2000 highly variable genes using scanpy for the scRNA-seq matrix in the second batch, and used the same set of genes for all the scRNA-seq matrices. We filled in pseudo-scRNA-seq matrix for the batches without scRNA-seq matrices, and selected the regions in scATAC-seq matrices that lie within the 2000 base-pair upstream or the gene body of the genes in scRNA-seq matrices. After the filtering process, we obtained overlapping 1677 genes and 25734 regions for all data matrices.

We download the cell label from the original data manuscripts, reorganize the labels to make them as consistent as possible. We re-annotate the "E2Rasgrf2", "E3Rmst" and "E3Rorb" as "L2/3", "E4Il1rapl2", "E4Thsd7a", "E5Galnt14", "E5Parm1", "E5Sulf1", and "E5Tshz2" as "L4/5", "E6Tle4" as "L6", "OliM" and "OliI" as "Oligo", "InV" as "CGE", "InS" as "Sst", "InP" as "Pvalb", "InN" as "Npy", and "Mic" as "MGC" in the first batch. We re-annotate "Lamp5", "Vip" and "Sncg" as "CGE", "L4", "L5 ET" and "L5 IT" as "L4/5", "L6 CT", "L6 IT" and "L6b" as "L6", "L5/6 NP" as "NP", "Macrophage" as "MGC" in the second batch. We re-annotate "L5.IT.a", "L5.IT.b" and "L4" as "L4/5", "L6.CT" and "L6.IT" as "L6", "L23.a", "L23.b", and "L23.c" as "L2/3", "OGC" as "Oligo", "ASC" as "Astro", and "Pv" as "Pvalb" in the third batch. We re-annotate "L2/3 IT" as "L2/3", "L4", "L5 IT", and "L5 PT" as "L4/5", "L6 CT", "L6 IT", and "L6b" as "L6", "Macrophage" as "MGC", "Lamp5", "Vip", and "Sncg" as "CGE" in the fourth and the fifth batches.

**Healthy human BMMC dataset.** We selected top 1000 highly variable genes using scanpy for scRNA-seq matrix. We also remove the genes

with no regions in the scATAC-seq matrices lying within the 2000 bp of their upstream sequence. The filtering process gives us 924 genes 22133 regions.

**Mouse spleen dataset.** We selected top 3000 highly variable genes using scanpy for the scRNA-seq matrix. We also remove the genes with no regions in the scATAC-seq matrices lying within the 2000 bp of their upstream sequence. The filtering process gives us 2708 genes 20435 regions.

### Running baseline methods
**UINMF and LIGER.** We followed its online tutorial (http://htmlpreview. github.io/?https://github.com/welch-lab/liger/blob/master/vignettes/ SNAREseq_walkthrough.html) to run UINMF. When applying UINMF to real datasets, we first binned the genome into bins of 100,000 bp for raw scATAC-seq matrices. We then used the same the pseudo-scRNA-seq and scRNA-seq matrices as the ones that were used in scMoMaT, but normalized the matrices following UINMF tutorial instead of performing quantile normalization and log transform. We select the latent dimension to be 30 as was recommended in the tutorial. When running on simulated dataset, We generated pseudo-scRNA-seq matrices following the same procedure as scMoMaT. Setting the number of latent dimension to 30 led to very bad results so we used the ground truth number of clusters as the number of latent dimensions. Since UINMF is extended from the LIGER framework, we ran LIGER with the same hyper-parameter setting as UINMF.

**MultiMap.** We ran MultiMap following the example in its GitHub repository (https://github.com/Teichlab/MultiMAP). We ran the method using the raw data matrices as was required by the example and generated the pseudo-scRNA-seq matrices following the same procedure as was used in scMoMaT. MultiMap can be directly applied to integration scenarios with no batch that has paired data, including the human BMMC dataset, mouse spleen dataset, and the first scenario of the simulated dataset, following the example. Where there exist batches with more than one modality profiled, including the human PBMC dataset and the second scenario of the simulated dataset, we concatenated the count matrices for each batch and calculated the low-dimensional representation using PCA on the concatenated matrices (as suggested by the authors). We ran MultiMap using the default hyper-parameter setting in its example.

**StabMap.** We ran StabMap following its online tutorial (https:// marionilab.github.io/StabMap/articles/stabMap_PBMC_Multiome. html). StabMap can be directly applied to the integration scenario where there exists paired data to connect different modalities. When running StabMap on the integration scenarios where no paired data exist to connect every modality, we generate pseudo-counts to connect different modalities instead. For a fair comparison result, we used the same set of pseudo-count matrices that were used in scMoMaT when running StabMap. In addition, we used the first batch as the reference batch, and used the default parameters setting when running the algorithm.

**Seurat.** We ran Seurat following its online tutorial (https://satijalab. org/seurat/articles/atacseq_integration_vignette.html), and set the hyper-parameter of Seurat to be exactly the same as its online tutorial.

### Evaluation metrics
**Graph connectivity.** Graph connectivity (GC) score measures how well the cells of the same cell type between batches are mixed in the latent space[21]. GC score is calculated by first constructing a kNN graph using cells from all batches. Then for each cell type, we select the cells that belong to the cell type and denote the corresponding subgraph by $G_c(N_c, E_c)$ where $c$ denotes the cell type. The GC score of this cell type can be calculated as $|LCC(G_c)|/|N_c|$, where $|LCC(G_c)|$ denotes the largest number of connected cells within the subgraph $G_c$, and $|N_c|$ denotes the total number of cells in the subgraph. The GC score of the whole dataset is the average GC score of all cell types.

**Adjusted Rand Index (ARI).** The ARI score measures how well cells from different cell types can be correctly clustered regardless of batches using the latent embedding. After clustering the cells using the cell latent embedding obtained from different integration methods, we calculate the Adjusted Rand Index[58] by comparing it with the ground truth cell label. Leiden clustering algorithm has one resolution parameter that decides the number of clusters. For each method, we ran Leiden clustering with different resolution parameters (from 0.1 to 10 with stepsize 0.5) and report the highest ARI score for all resolution parameters as the final result.

**Normalized mutual information (NMI) score.** Similar to ARI score, NMI score also measures how well cells from different cell types can be correctly clustered using the latent embedding. NMI is calculated with both the cluster label and ground-truth label. For each method, we obtained the cluster label using Leiden clustering algorithm, ran the clustering algorithm with different resolution parameters (from 0.1 to 10 with stepsize 0.5) and report the highest NMI score for all resolution parameters.

**k-Nearest neighbor (kNN) agreement score.** The kNN agreement score is designed to evaluate a set of cell-type annotations that are often obtained on integrated datasets. Ideally, a set of cell type labels obtained on the integrated dataset should also be able to separate cells in each individual data matrix before integration. To quantify to what extent a set of labels "separate" the cells in each data matrix, we calculate the kNN agreement score for each cell in this data matrix. Given the cell type labels for a set of cells, the kNN agreement score measures the label agreement between each cell and its nearest neighbors. Intuitively, with high-quality labels, cells with different labels should be separated, so most of the cells should have the same label as their neighbors, unless the cell is at the boundary of two or more closely located clusters. Taking a scRNA-seq matrix as an example: we first construct a kNN graph of cells using pairwise distances obtained after performing PCA on the original data matrix (using the top 30 principle components). For each cell, we calculate the proportion of cells that share the same label with this cell in its $k$ nearest neighbors, and the kNN agreement score of a dataset is the average of this proportion over all cells. The procedure is the same for protein abundance matrices. For scATAC-seq matrices, latent semantic indexing (LSI, using the top 30 components) was used to reduce the dimension of the original matrices in order to construct the kNN graphs.

**F1 score.** F1 score is used for rare cell type detection[59]. Given the ground truth and predicted rare cell types, F1 score is calculated as the harmonic mean of precision and recall:

$$F_1 = 2 \times \frac{\text{precision} \times \text{recall}}{\text{precision} + \text{recall}} \tag{7}$$

We assigned the predicted rare cell type to be the cell cluster that has the largest overlap with the ground truth rare cell type.

### Statistics and reproducibility
The datasets in this study are previously published, and the sample sizes are pre-determined in the original datasets. No data were excluded from the analyses results. The study does not involve the use of experimental replication, randomization, and allocation blinding.

## Reporting summary

Further information on research design is available in the Nature Portfolio Reporting Summary linked to this article.

## Data availability

All datasets used in this paper are previously published and freely available. The human PBMC dataset is available at Gene Expression Omnibus under accession number GSE156478. The first batch in the mouse brain cortex dataset can be accessed at Gene Expression Omnibus under accession number GSE126074. The second and the third batches are from Yao et al. and accessed at NeMO Archive with accession number nemo:dat-ch1nqb7[36]. The fourth batch is from Allen Brain Atlas [http://celltypes.brain-map.org/api/v2/well_known_file_download/694413985][37,38]. The fifth batch is from 10x Genomics website[60] [https://support.10xgenomics.com/single cell-atac/datasets/1.1.0/atac_v1_adult_brain_fresh_5k]. The healthy human BMMC dataset is available at Gene Expression Omnibus under accession number GSE139369. The scATAC-seq and scRNA-seq matrix of mouse spleen dataset are available at ArrayExpress under accession numbers E-MTAB-6714 and E-MTAB-9769. All other relevant data supporting the key findings of this study are available within the article and its Supplementary Information files or from the corresponding author upon reasonable request. Source data are available through Zenodo[61]. Source data are provided with this paper.

## Code availability

scMoMaT is implemented as a python package (python ver. 3.8.10, pytorch ver. 1.11.0) that is available at https://github.com/PeterZZQ/scMoMaT. The package version used for the analyses in the paper has been assigned a citable DOI through Zenodo https://doi.org/10.5281/zenodo.7523552.

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

## Acknowledgements

This work was supported by the US National Science Foundation DBI-2019771 and National Institutes of Health grant R35GM143070 (Z.Z., X.Z.), the Wellcome Sanger core grant WT206194 (S.A.T.), Singapore Ministry of Education Academic Research Fund R-253-000-138-133 (V.R.), Shenzhen Innovation Committee of Science and Technology 202208150943330001 (Xi C.) and a Gates Cambridge Scholarship (M.S.J.).

## Author contributions

Z.Z., R.M., V.R. and X.Z. conceived the idea of scMoMaT. Z.Z., H.S. and Xinyu C. designed and implemented the scMoMaT algorithm. Z.Z. carried out the evaluation and data analysis. M.S.J., M.E., S.A.T. and Xi C. helped with the data analysis. Z.Z. and X.Z. wrote the paper.

## Competing interests

In the past three years, S.A.T. has consulted for or been a member of scientific advisory boards at Qiagen, Sanofi, GlaxoSmithKline and ForeSite Labs, and is a consultant and equity holder of TransitionBio. The other authors declare no competing interests.
