## [Peer Review File · Nature Communications]

REVIEWER COMMENTS

Reviewer #1 (Remarks to the Author):

The authors develop a method called scMoMaT for integrative analysis of single cell genomics data from multiple modalities and multiple batches. Such integrative analysis task is called mosaic integration, and it is the most general and challenging tasks in integrative analysis. The Matrix Trifactorization-based method presented in this manuscript is a general approach and technically solid. However, I have some concerns about the manuscript.

1. In introduction, authors state that previous mosaic integration methods focus on learning cell representations while the scMoMaT also aims to identify cell type specific biomarkers across modalities. Most research did/will do this in a two-steps-approach, identify cell population first, and identify biomarkers by a differential testing. It is not clear that what is the advantage of retraining of scMoMaT compared to those simple statistical testing? Authors should compare the biomarkers identified by scMoMaT and those from differential testing to illustrate the advantage of the method.
2. In the PBMC data section, the ground truth cell label is from the ref #3, which is the original paper of ASAP-seq. The ground truth of the CITE-seq data is not discussed. It is not clear how authors validate the results without using the label of CITE-seq data.
3. The KNN agreement used for comparison of scMoMaT and other methods is not meaningful because the distance is calculated on the Umap space which is from the cell representation matrix from scMoMaT. Thus, scMoMaT is expected to perform better for this metric than other methods. If the Umap is based on UINMF or MultiMap, the results could be different.
4. In mouse cortex data, scMoMaT took as input pseudo-scRNAseq (gene activity score?) from scATAC-seq data. What the results look like if authors do not take the gene activity score as input? I think the method still should work since the first batch data contain both gene expression and chromatin accessibility data. I believe author do this is because the method performance was improved by using the gene activity score as inputs. However, it is also worth to show that the method works good even without gene activity score.
5. In terms of NMI and ARI, scMoMaT has similar performance with UINMF on diagonal integration. What is the performance of the method compared to other diagonal integration methods such as Seurat v3 and LIGER?

Reviewer #2 (Remarks to the Author):

Zhang et al. proposes a method names scMoMaT for mosaic integration of single-cell data. Mosaic integration is a challenging problem for single-cell studies, so this is an important topic. Even though I found the topic of this manuscript interesting, there are several major concerns that need to be addressed to determine the actual applicability of the proposed method and if it provides advantages over existing methods. Additional computational evidences are also in need to justify the accuracy and robustness of the method.

1. Section 2.1 gives the mathematical formulation of scMoMaT. However, several key steps are provided without justifications, making it difficult to understand the objective function. For example, what are the dimensions of the cell factors and feature factors? What do the bias vectors represent? What's the purpose of the scaling parameters? What are the meanings of values in the association matrix?
2. The manuscript states that "When there is no common modality across batches (Fig. S1b), we add pseudo-count matrices to make the corresponding modality shared across all batches." However, this procedure described in Section 5.2 assumes scATAC-seq data is available and one of the available data modalities is gene expression. This limits the applicability of scMoMaT.

3. The complete algorithm for updating the cell and feature factors and association matrices should be provided to increase reproducibility of the work. Currently, only solutions for the bias terms and scaling factors are given.

4. scMoMaT has multiple parameters, but the manuscript does not explain how these parameters were selected for the simulation and real data analyses, and how they should be selected in real practice. How does it affect the performance if the parameters used in current analyses are altered?

5. The manuscript presents a simulation study to test scMoMaT's performance in integrating scRNA-seq and scATAC-seq data. Since many methods have been proposed for joint analysis of scRNA-seq and scATAC-seq data, while scMoMaT is proposed as a general method for multiomics integration, it would be more interesting to see evaluations based on more than 2 modalities, which are more challenging.

5. What are the cell numbers in the simulated data? It would be necessary to see performance of the method when cell numbers are not balanced. What happens when there are rare cell types?

6. The manuscript mentioned StabMap as a previous method for mosaic integration, but the computational comparisons only included two other methods and did not include StabMap.

7. Why comparisons with existing methods are not provided for the mouse cortex data? The current results do not support an obvious advantage of scMoMaT over UINMF, except on the PBMC dataset.

8. Horizontal and diagonal integrations are special cases of mosaic integration and are very common in single-cell studies. One question the manuscript does not discuss is that whether scMoMaT outperforms existing methods for these two special cases? Or it's only supposed to be used in strict mosaic integration problems?

Minor:

The feature embedding plot in Figure 1 is not legible.

Responses to Reviewers' Comments

We thank the reviewers for their time to review our manuscript and their valuable comments.

Please find below the reviewers' comments (in black) and our pointwise responses (in blue). The changes in the manuscript are marked in red. We number all the figures in this document by Fig. R1, Fig. R2, Fig. R3, etc, to distinguish them from the figures in the manuscript.

Ziqi Zhang, Haoran Sun, Ragnathan Mariappan, Xi Chen, Xinyu Chen, Mika S Jain, Mirjana Efremova, Sarah A Teichmann, Vaibhav Rajan, Xiuwei Zhang

Reviewer #1

The authors develop a method called scMoMaT for integrative analysis of single cell genomics data from multiple modalities and multiple batches. Such integrative analysis task is called mosaic integration, and it is the most general and challenging tasks in integrative analysis. The Matrix Trifactorization-based method presented in this manuscript is a general approach and technically solid. However, I have some concerns about the manuscript.

1. In introduction, authors state that previous mosaic integration methods focus on learning cell representations while the scMoMAT also aims to identify cell type specific biomarkers across modalities. Most research did/will do this in a two-steps-approach, identify cell population first, and identify biomarkers by a differential testing. It is not clear that what is the advantage of retraining of scMoMaT compared to those simple statistical testing? Authors should compare the biomarkers identified by scMoMaT and those from differential testing to illustrate the advantage of the method.

According to the reviewer's suggestion, we included the tests on the detection accuracy of cluster-specific biomarkers, and specifically, the differentially expressed genes between one cluster and the rest of the cells. Since the real datasets do not have ground truth biomarkers for comparison, we test the detection accuracy of scMoMaT with simulated datasets. The SymSim package allows users to output ground truth differentially expressed genes between two groups of cells based on mRNA counts without intrinsic and technical noise. The ground truth is in the form of a ranking of genes on how likely a gene is a DE gene.

We used the 8 simulated datasets described in the main manuscript, and extracted their ground truth cluster-specific genes from the simulator. As for the baseline method which performs the two-step process (first clustering and then performing DE), we use UINMF to perform integration and clustering. Then, based on the clustering results, we performed DE detection using Wilcoxon rank sum test which is considered to be one of the top performers for DE gene detection (Soneson and Robinson, 2018; Zhang, Xu and Yosef, 2019). We ran Leiden clustering

algorithm to obtain the cluster assignment from scMoMaT and UINMF. We selected the resolution parameter for the algorithm to the resolution that produces the best NMI score for each method. The two-step process outputs p-value using which we can rank genes on how likely a gene is a DE gene. scMoMaT outputs a gene score for each gene which also gives us a ranking of the genes.

We measured the DE detection accuracy of scMoMaT and baseline two-step procedure using Kendall-tau coefficient between the ranking of each method and the ground truth ranking. We summarized the performance of scMoMaT and baseline methods in the barplot below, where scMoMaT shows a consistently better performance compared to the baseline method (labeled as “Wilcoxon” in the plot). The better detection accuracy of scMoMaT is likely due to that scMoMaT jointly clustered the cells and the features at the same time, which improves both the cell cluster identification and marker gene detection accuracy at the same time.

We included these results as Fig. S3a and modified the manuscript to include the description and analysis of this additional test (Page 19, Line 525-532; Page 7, Lines 166-177).

2. In the PBMC data section, the ground truth cell label is from the ref #3, which is the original paper of ASAP-seq. The ground truth of the CITE-seq data is not discussed. It is not clear how authors validate the results without using the label of CITE-seq data.

Both the ASAP-seq (batches 3 and 4 in our Fig. 3a) and CITE-seq (batches 1 and 2 in our Fig. 3a) data we used in Section 2.3 are from ref #3 (Mimitou *et al*). The authors provided cell-type annotations for both the ASAP-seq and the CITE-seq data, and we used the annotated labels as the ground truth in our test. In the manuscript, we have modified Page 8, Lines 200-201 to make this clear.

3. The KNN agreement used for comparison of scMoMaT and other methods is not meaningful because the distance is calculated on the Umap space which is from the cell representation matrix from scMoMaT. Thus, scMoMaT is expected to perform better for this metric than other methods. If the Umap is based on UiNMF or MultiMap, the results could be different.

We thank the reviewer for this question. We would like to clarify the following points:

1. The kNN graphs are not constructed based on the UMAP space. The UMAP shown in Fig. S5 is only for visualization.

2. The kNN graphs are constructed after performing PCA (for scRNA-seq data matrices or protein data matrices, using the top 30 principle components) or latent semantic indexing (LSI, for scATAC-seq data matrices, using the top 30 components) on each data matrix before applying any integration methods. Therefore, the kNN graphs are independent of any integration method (scMoMaT, UiNMF, or MultiMap).
3. Fig. S5 shows the UMAP of each data matrix before applying any integration method, and the cells are colored with cell labels derived from scMoMaT (top panel) or from the original publication (bottom panel). We would like to show that the scMoMaT labels further improve the original labels in terms of the kNN agreement score. The kNN agreement score is motivated by the following intuition: ideal cell type labels (obtained on the integrated dataset) should be able to separate cells when considering each data matrix separately. Visually, cases like the areas in red circles/ovals in Fig. S5 should be as few as possible.
4. Thus, for each data matrix before integration, we first construct a kNN graph, and then for each cell, we calculate the proportion of cells in its neighborhood that share the same cell type as the central cell. This proportion is calculated for all cells in all input data matrices, and an average is obtained in the end as the kNN agreement score of an annotation.

We have modified the manuscript to better convey the process of calculating the kNN agreement scores (Page 9, Lines 215; Page 22-23, Lines 631-643).

4. In mouse cortex data, scMoMaT took as input pseudo-scRNAseq (gene activity score?) from scATAC-seq data. What the results look like if authors do not take the gene activity score as input? I think the method still should work since the first batch data contain both gene expression and chromatin accessibility data. I believe author do this is because the method performance was improved by using the gene activity score as input. However, it is also worth showing that the method works well even without gene activity scores.

We thank the reviewer for the suggestion. As the reviewer mentioned, the model still works without the pseudo-scRNA-seq matrices (which are named gene activity scores in some literature), since the first batch contains the information that connects the two modalities. The main reason we include the pseudo-scRNA-seq matrices for the mouse cortex dataset is that the scATAC-seq data in the first batch (measured by SNARE-Seq) has much higher sparsity compared to single-modality scATAC-seq data. This is also reflected in Fig. S10 (top-right plot), where the cell types are less separated compared to visualizations from other data matrices in Fig. S10. The low quality of the scATAC-seq data matrix in the SNARE-Seq data batch makes the connection between the two modalities in this batch relatively weak.

We have conducted additional tests to show the following points:

1. With the five batches of mouse cortex data, without any pseudo-scRNA-seq data the performance has decreased (Fig. R1 below). We hypothesize that this is due to the low quality of the scATAC-seq data matrix in the SNARE-seq dataset. The results of the test described in point #3 support this hypothesis.

- Using only one pseudo-scRNA-seq data matrix, scMoMaT is able to perform the integration well (Fig. R2 and Fig. R3 below).
- We show that if the quality of the scATAC-seq data matrix is improved, then scMoMaT can perform well without using any pseudo-scRNA-Seq. We show this with simulated data where we can control the quality of the scATAC-seq data matrix (Fig. R4 below).

Note that other mosaic integration methods, UINMF and MultiMap, both require pseudo-scRNA-seq data matrices for all batches with only scRNA-seq data. When applying scMoMaT in practice, we suggest including one or more pseudo-scRNA-seq matrices in case the quality of scATAC-seq data matrix affects the integration.

Figure R1. The UMAP visualization of cell factors in scMoMaT, where all pseudo-scRNA-seq are removed. In these results, the cell types in batches 3 and 5 (single-modality scATAC-seq) are correctly aligned, and cell types in batches 1, 2, 4 (scRNA-seq) are also correctly aligned. However, cell types from batches 3, 5 are not correctly aligned with batches 1, 2, 4. The result shows that scMoMaT learns to align all batches separately for each modality, but the connection between modalities, which is only provided by batch1, is not strong enough to be learned.

Figure R2. The UMAP visualization of cell factors in scMoMaT, where only one pseudo-scRNA-seq data matrix (batch 3) is used. This time all batches are correctly aligned, even for batch 5 which is only measured with scATAC-seq. This is because stronger cross-modality connection information is provided through the pseudo-scRNA-seq data matrix.

Figure R3. The UMAP visualization of cell factors in scMoMaT, where only one pseudo-scRNA-seq data matrix (batch 5) is used. This time all batches are correctly aligned, even for batch 3 which is only measured with scATAC-seq. This is because stronger cross-modality connection information is provided through the pseudo-scRNA-seq data matrix.

	Original	More drop-out
GC	0.985	0.797
NMI	0.794	0.563
ARI	0.645	0.350

Fig R4. The UMAP visualization cell factors for the simulated dataset, and the corresponding benchmarking scores. Here we conduct experiments on how the quality of jointly sequenced data can affect the integration accuracy of scMoMaT. We generated one simulated dataset with three batches, where batch 1 is measured with only scATAC-seq, batch 2 is measured with only scRNA-seq, and batch 3 is measured with both scRNA-seq and scATAC-seq. Each batch can have some missing cell types. We created two integration scenarios with different data qualities of batch 3. In the first scenario, we used the original simulated data where the data quality is considered good. In the second scenario, we add additional drop-outs into the scATAC-seq data of batch 3 to worsen the data quality. We ran scMoMaT on the simulated dataset without using any pseudo-scRNA-seq count matrices. scMoMaT correctly matches cells of different cell types in scenario 1, while it cannot match cells of different cell types in scenario 2. The benchmark scores (GC, NMI, ARI) also show that the performance of scMoMaT is affected by the quality of jointly sequenced data batches that are used to connect different modalities. Since some of the joint sequencing methods can have low sequencing quality for each modality, we recommend the users use the pseudo-scRNA-seq data when conducting integration.

5. In terms of NMI and ARI, scMoMaT has similar performance with UINMF on diagonal integration. What is the performance of the method compared to other diagonal integration methods such as Seurat v3 and LIGER?

We thank the reviewer for this question. We now have also run Seurat v3, LIGER, and StabMap (as suggested by Reviewer #2) on the two diagonal datasets. The new results are in Fig. 6d,

S12 (for the human bone marrow dataset) and Fig. 6i, S15, S16 (for the mouse spleen dataset). On the human bone marrow dataset, scMoMaT and Seurat v3 are two top performers with comparable performances. On the mouse spleen dataset, scMoMaT has the best overall performance.

We modified the manuscript accordingly:

1. We updated the text to reflect the additional results (Pages 12: Lines 338 - 343, Pages 13-14: Lines 376-383).
2. We have added details on how we ran the additional baseline methods, Seurat v3, LIGER, and StabMap in Methods (Pages 21-22 Lines 579-612).

Reviewer #2

Zhang et al. proposes a method names scMoMaT for mosaic integration of single-cell data. Mosaic integration is a challenging problem for single-cell studies, so this is an important topic. Even though I found the topic of this manuscript interesting, there are several major concerns that need to be addressed to determine the actual applicability of the proposed method and if it provides advantages over existing methods. Additional computational evidences are also in need to justify the accuracy and robustness of the method.

1. Section 2.1 gives the mathematical formulation of scMoMaT. However, several key steps are provided without justifications, making it difficult to understand the objective function. For example, what are the dimensions of the cell factors and feature factors? What do the bias vectors represent? What's the purpose of the scaling parameters? What are the meanings of values in the association matrix?

We thank the reviewer for pointing out the need for further explanation on the mathematical formulation of scMoMaT. We have modified the manuscript to improve the presentation of Section 2.1 *Framework of scMoMaT* (Pages 3-5), and we also summarize the answers to the specific questions in this comment in the following:

1. Regarding the dimensions of the *cell factors* and *feature factors*: suppose the input data matrix has n cells and m features (a feature is a gene if the data matrix is scRNA-seq data), and the number of latent dimensions is d , then the cell factor of this input matrix is of size $n \times d$ and the feature factor of this input matrix is of size $m \times d$. d is usually a value much smaller than the number of cells or features of the input count matrix (eg. d is set to 30 in all real datasets in the manuscript). Each row of the cell factor is the latent presentation of a cell and each row of the feature factor corresponds to the latent representation of a feature (eg. a gene).
2. Regarding the *cell bias* and *feature bias* terms: the cell bias term represents the data-matrix-specific variation among cells, and the feature bias represents the data-matrix-specific variation among features. Taking an input scRNA-seq count matrix

X_{ij} as an example. Rows of this matrix correspond to cells from the i^{th} batch and columns correspond to the j^{th} modality. According to Fig. 1b, X_{ij} is tri-factorized into three matrices: the cell factor C_i , the association matrix Σ_{ij} , and the gene factor C_j . C_i and C_j can be shared with other input data matrices, eg., another scRNA-seq count matrix shares the same C_j . Since C_i and C_j are shared between data matrices, the cell bias and feature bias terms are introduced to represent data-matrix-specific information.

3. The *scaling parameters* are introduced because count matrices of different modalities may not have counts on the same scale (some data matrices may have larger average count values than others).
4. The *association matrix* encodes the interaction between cell factors and feature factors. The value in row r and column c of the association matrix Σ_{ij} means the interaction strength between the r^{th} latent dimension of the cell factor and c^{th} latent dimension of the feature factor.

The cell bias, feature bias, scaling parameter, and association matrices are all learned together with the cell factors and feature factors.

2. The manuscript states that “When there is no common modality across batches (Fig. S1b), we add pseudo-count matrices to make the corresponding modality shared across all batches.” However, this procedure described in Section 5.2 assumes scATAC-seq data is available and one of the available data modalities is gene expression. This limits the applicability of scMoMaT.

We thank the reviewer for pointing this out. We realize that our previous statements on Page 5 and Fig. S1 are misleading and do not accurately describe the application scenarios of scMoMaT. We now have modified Section 2.1 (Page 5) and Fig. S1 to provide a more detailed and accurate description of the application scenarios. We also have added procedures to calculate pseudo-count matrices in the following modes: (1) Given gene expression, calculate pseudo-count-matrix of protein abundance; (2) Given chromatin accessibility, calculate pseudo-count-matrix of protein abundance.

With these modifications, the points we would like to clarify are the following:

1. scMoMaT does not require there exists at least one modality that is shared across all batches. Instead, scMoMaT only requires that the entities of data matrices are connected (further explanations are given on Page 5, Lines 91-102). For example, the scenario in Fig. S3b does not require any pseudo-count matrix.
2. Practically, however, adding pseudo-count matrices when they are not theoretically required can lead to better integration by providing stronger cross-modality relationships. Relevant results can be seen in Fig. S3. Baseline methods also theoretically require pseudo-count-matrices (UINMF, MultiMap, LIGER, and Seurat) or recommend using pseudo-count-matrices. More detailed discussions on the role of pseudo-count-matrices for scMoMaT can be found in our response to Question #4 of Reviewer #1.

3. Pseudo-count matrices can be calculated for the diagonal cases of any two modalities out of the three most commonly used modalities: gene expression, chromatin accessibility, and protein abundance. We have updated Section 5.2 with procedures for calculating the pseudo-protein-count matrices from either scRNA-seq matrix or scATAC-seq matrix. These pseudo-protein-count matrices are shown to work in Fig. R5 (the responses to Question #5 of Reviewer #2).

3. The complete algorithm for updating the cell and feature factors and association matrices should be provided to increase reproducibility of the work. Currently, only solutions for the bias terms and scaling factors are given.

We thank the reviewer for pointing this out. We now have added the pseudocode of the complete algorithm in the Supplementary Material (*Supplementary Note 1: pseudo-code of scMoMaT*). When updating the cell factors, feature factors, and association matrices with stochastic gradient descent, the gradients of the parameters are calculated automatically with *PyTorch*.

4. scMoMaT has multiple parameters, but the manuscript does not explain how these parameters were selected for the simulation and real data analyses, and how they should be selected in real practice. How does it affect the performance if the parameters used in current analyses are altered?

We thank the reviewer for this suggestion. There are four hyper-parameters in scMoMaT: the latent dimension d for cell and feature factors, the regularization weight λ in the loss function, the number of neighbors k , and radius parameter r in the post-processing step. We have added a subsection in the manuscript (Page 14, Section 2.7, *Effects of hyper-parameters settings in scMoMaT*), which points to our discussion in Methods that include the parameter setting for the results presented in the manuscript, as well as guidance for users on how to set these parameters (Page 18, Section 5.5).

To answer how the scMoMaT performance is affected by altered parameter settings, we performed additional tests with varying hyperparameter values, using the simulated datasets illustrated in Fig. 2b. We observe how the three metrics, NMI, ARI, and GC change when changing hyper-parameters.

We first vary the two parameters used in the training stage, d and λ , where $d = 10, 20, 30, 50$; and $\lambda = 1e-4, 1e-3, 1e-2$. The results are summarized in barplots below. We can see that all three metrics are not sensitive to the change of λ , except for the case where $d=10$, which is a number too small for the latent dimensions. The results are also robust with the change in d , as long as that $d \geq 20$.

We then vary the two parameters used in post-processing stage k and r , where $k = 15, 30, 50$; and $r = 0.7, 0.9, 1$. The results are summarized in barplots below. First, we can see that the differences in the metrics when changing k are very small. In the case of r , we see that the ARI and NMI decreases when increasing r , although only by a small amount. This is because these datasets have unmatched cell types across data batches, and applying a smaller r is suggested for the case with unmatched cell types. When the datasets are expected to have the same cell types, we suggest setting $r=1$ to skip the pruning step (more details in Methods in the manuscript).

Overall, these results show that scMoMaT is robust to the hyper-parameter setting. These figures are included in Fig. S17. Discussions on these new results are included in Page 14, Section 2.7, *Effects of hyper-parameters settings in scMoMaT*.

5. The manuscript presents a simulation study to test scMoMaT's performance in integrating scRNA-seq and scATAC-seq data. Since many method have been proposed for joint analysis of scRNA-seq and scATAC-seq data, while scMoMaT is proposed as a general method for multiomics integration, it would be more interesting to see evaluations based on more than 2 modalities, which are more challenging.

We thank the reviewer for this suggestion on highlighting one of the scMoMaT's advantages with simulated datasets. We have expanded our data simulation procedure to also simulate the protein abundance modality, in addition to the existing two modalities: gene expression and chromatin accessibility. The simulation procedure is as follows:

1. We first select a reference protein count dataset and fit its counts into a *protein count distribution*. We then can sample new protein counts from this distribution. We use the protein count in the human PBMC dataset (from (Mimitou *et al.*, 2021), the first real dataset in the manuscript) as the reference dataset.
2. For each batch in the simulated datasets, we generated its protein count matrix from the corresponding gene expression matrix. As current protein abundance data matrices profile only a subset of proteins, we select the top 200 highly variable genes from the

gene expression matrix, and generate counts for the corresponding 200 proteins, where each protein is associated with one gene.

3. We assume the protein counts are positively correlated with the corresponding gene counts, and are sampled from the reference *protein count distribution*. We draw $200 \times n$ samples from the distribution, where n is the number of cells in the given batch. We fill these $200 \times n$ samples into the protein count matrix according to the rank of their corresponding gene expression level. To mimic that in real data we do not observe a perfect correlation between a gene's expression and its corresponding protein abundance, we randomly select 10% entries in the protein abundance matrix, and randomly permute their rank-based value assignment when filling them into the protein count matrix.

After generating the protein count for each batch in the simulated dataset, we test scMoMaT and baseline methods using the following scenario:

Note that both MultiMap and UINMF require that there exists a modality that is available in all batches. So we generated pseudo-count matrices for the protein modality for batches 1, 2, 5, and 6 shown in the following figure. For batches 1 and 2, the pseudo-protein count matrices are generated from the scATAC-seq modality, and for batches 5 and 6, the pseudo-protein count matrices are generated from the scRNA-seq modality. scMoMaT does not require pseudo-protein count matrices to be provided in this scenario, and we show the performance of scMoMaT with and without pseudo-protein count matrices.

We compare the performance of scMoMaT and baseline methods on 8 simulated datasets (corresponding to 8 random seeds). scMoMaT was run with two modes: with or without pseudo-protein-count matrices. The performance comparisons are shown in the boxplots below:

Fig R5. The benchmarking scores of scMoMaT (using pseudo-count matrix and not using pseudo-count matrix) and baseline methods on the simulated datasets with three modalities.

scMoMaT consistently shows a better performance compared to the baseline methods, and the inclusion of pseudo-protein count matrices further improves the performance of scMoMaT. These results also illustrate the use of pseudo-protein-count matrices from scATAC-seq and scRNA-seq matrices, thus helping to address Question #2 from Reviewer #2.

We also modified the main manuscript accordingly to include the test results above (Pages 7-8, Lines 178-185, section 2.2.2, Fig. S3).

These additional results on simulated datasets and the results on the first real dataset (the human PBMC dataset which includes cells measured with three modalities: protein abundance, gene expression, and chromatin accessibility) in the manuscript together validate the performance of scMoMaT when integrating data matrices from three modalities.

6. What are the cell numbers in the simulated data? It would be necessary to see performance of the method when cell numbers are not balanced. What happens when there are rare cell types?

The answer to each question is stated separately below:

(1) There are 8 simulated datasets for the simulation tests in the manuscript. For both scenarios shown in Fig. 2a and Fig. 2b, there are 6 batches generated for each dataset, which includes 10,000 cells in total. The number of cells in each batch is around 1,600, with a small deviation. There are 16 cell types in each dataset, including 15 normal cell types and 1 rare cell type. Each normal cell type includes 660 cells across 6 batches, and the rare cell type includes 100 cells across 6 batches ($660 \cdot 15 + 100 = 10,000$).

After the 10,000 cells are simulated, we further randomly downsample the cell population to account for the unequal cell type composition across batches (lines 122-124 on page 6). After downsampling, the number of cells in each batch is around 1,100 with a small deviation. The total number of cells in a simulated dataset is around 7,000 including 6 batches. The number of cells for each normal cell type is around 400, and for the rare cell type is around 60. We add the

detailed simulation parameter setting (number of genes, regions, proteins, and the number of cells in each batch of each dataset) into the manuscript (Table S1).

(2) According to the reviewer's suggestion, we have tested scMoMaT on simulated datasets where the cell numbers are imbalanced across batches. We create the imbalanced dataset by subsampling the cells of 3 batches in the original simulated datasets. For cell batches 1, 4, and 6 in each dataset, we subsampled their cells such that each of these three batches has a number of cells that is a tenth of each of the remaining batches. We run scMoMaT on the subsampled datasets, and compare them with the baseline methods. The comparison result is shown below:

(2). As described in (1), each batch has one rare cell type. Here we test scMoMaT and baseline methods on the current simulated datasets, and measure their rare cell type detection accuracy using F1 score. The same metric was also used for rare cell type detection in existing work (Fa *et al.*, 2021). The results are shown below:

scMoMaT has the highest F1 score compared to baseline methods.

7. The manuscript mentioned StabMap as a previous method for mosaic integration, but the computational comparisons only included two other methods and did not include StabMap.

According to the reviewer's suggestion, we added the comparison result with StabMap on both real and simulated datasets in the main manuscript. Please refer to *Sections 2.2, 2.3, 2.5, 2.6*, and the corresponding *Figs. 2, 3, 6, S3, S4, S12, S15, and S16* for the comparison between scMoMaT and StabMap. scMoMaT outperforms StabMap in all these results.

8. Why comparisons with existing methods are not provided for the mouse cortex data? The current results do not support an obvious advantage of scMoMaT over UINMF, except on the PBMC dataset.

The data batches of the mouse cortex dataset are gathered from different publications where the cell type labeling is highly inconsistent. For example, cell types Clau, Mis, and Npy are only in batch 1; Smc is only in batches 2 and 4; CR and Serpinf1 are only in batch 4; Meis2 and VLMC are only in batches 4 and 5. Therefore, we did not find cell-type labels that are reliable enough to serve as ground truth labels for quantitative evaluation of the integration methods. So we instead annotated the cell types using scMoMaT, and indirectly validated the annotations using biological information obtained from the existing literature (marker genes and motifs, Fig. 4b,e,d; Fig. S11; Fig. 5).

The major advantages of scMoMaT over UINMF are as follows:

(1) In terms of integration performance measured on the integrated cell representations (the GC, ARI, NMI scores), scMoMaT outperforms UINMF on all scenarios with simulated data. With real datasets, scMoMaT outperforms UINMF on all three real datasets where ground truth cell labels are available (Fig. 3d, Fig. 6d, i). scMoMaT also obtained better results than UINMF in the horizontal integration case in the response to the next question (Question #9).

(2) scMoMaT not only learns the cell representations, but also the representations of features from every input modality, as well as the bio-markers from each modality for every cluster. For example, with the PBMC dataset, scMoMaT outputs marker genes, marker proteins, and marker motifs for every cluster to help with cell type annotations. UINMF outputs cell representations and cell clusters but not bio-markers.

(3) UINMF requires that there exists at least one modality that is available in all cell batches, and scMoMaT does not have this requirement.

9. Horizontal and diagonal integrations are special cases of mosaic integration and are very common in single-cell studies. One question the manuscript does not discuss is that whether scMoMaT outperforms existing methods for these two special cases? Or it's only supposed to be used in strict mosaic integration problems?

scMoMaT is applicable to horizontal, vertical, diagonal, and general mosaic integration scenarios. We tested scMoMaT on human bone marrow (Section 2.5) and mouse spleen datasets (Section 2.6), which both fall into the category of diagonal integration. scMoMaT achieves top performance on these two datasets (Fig. 6d, i).

To test the performance of scMoMaT under the horizontal integration scenario, we used a human pancreas dataset (Luecken *et al.*, 2021). The dataset includes the scRNA-seq data of 14890 cells in 8 batches. We ran scMoMaT, UINMF, and MultiMap on this dataset. The cell factor of scMoMaT is shown in Figs. R6a,b below, where cells of different cell types are separated and different batches are aligned. We quantitatively measure the integration accuracy using GC, ARI, and NMI scores (Fig. R6c). The scores show that scMoMaT outperforms UINMF with all scores; and compared to MultiMap, scMoMaT has a lower GC score but much higher NMI and ARI scores. Although MultiMap has good performance on this dataset, its

performance is not stable on different datasets (eg. the metric values are very low with the BMMC dataset in the paper).

Considering the length of the manuscript, we did not add the results on horizontal integration to the manuscript. But we are open to including them if the reviewers think these results are essential.

Figure R6. The test result of scMoMaT on human pancreas dataset. **a-b.** The cell factor of scMoMaT, visualized with UMAP. Cells are colored by (a) cell types and (b) cell batches. **c.** The scores of scMoMaT and baseline methods.

Overall, scMoMaT is applicable to the most general form of mosaic integration, and in terms of specific scenarios (diagonal integration, horizontal integration) it provides better or comparable performance compared to the state-of-the-art methods. Furthermore, scMoMaT simultaneously learns bio-markers from each input modality, which can include marker genes, marker proteins, and marker motifs to assist biological analysis of cell types obtained on the integrated data.

Minor:

The feature embedding plot in Figure 1 is not legible.
We have updated Fig. 1 to make it more clear.

References

Fa, B. *et al.* (2021) 'GapClust is a light-weight approach distinguishing rare cells from voluminous single cell expression profiles', *Nature communications*, 12(1), p. 4197.

Luecken, M.D. *et al.* (2021) 'Benchmarking atlas-level data integration in single-cell genomics', *Nature methods* [Preprint]. Available at: <https://doi.org/10.1038/s41592-021-01336-8>.

Mimitou, E.P. *et al.* (2021) 'Scalable, multimodal profiling of chromatin accessibility, gene expression and protein levels in single cells', *Nature biotechnology* [Preprint]. Available at: <https://doi.org/10.1038/s41587-021-00927-2>.

Soneson, C. and Robinson, M.D. (2018) 'Bias, robustness and scalability in single-cell differential expression analysis', *Nature methods*, 15(4), p. 255.

Zhang, X., Xu, C. and Yosef, N. (2019) 'Simulating multiple faceted variability in single cell RNA sequencing', *Nature communications*, 10(1), p. 2611.

REVIEWER COMMENTS

Reviewer #1 (Remarks to the Author):

Authors have addressed all my comments. I have no further comments.

Reviewer #2 (Remarks to the Author):

The revised manuscript has addressed most of my previous comments, and I only have one remaining concern: regarding the effect of hyper-parameters, it would be important to see the results on real data, in addition to simplified simulated data.

Responses to Reviewers' Comments

We highly appreciate the reviewers providing valuable feedback on our revised manuscript within a short time frame.

Please find below the reviewers' comments (in black) and our pointwise responses (in blue). The changes in the manuscript are marked in red.

Ziqi Zhang, Haoran Sun, Ragunathan Mariappan, Xi Chen, Xinyu Chen, Mika S Jain, Mirjana Efremova, Sarah A Teichmann, Vaibhav Rajan, Xiuwei Zhang

Reviewer #1

Authors have addressed all my comments. I have no further comments.

Reviewer #2

The revised manuscript has addressed most of my previous comments, and I only have one remaining concern: regarding the effect of hyper-parameters, it would be important to see the results on real data, in addition to simplified simulated data.

We thank the reviewer for this suggestion. We have conducted hyper-parameter tests on the human PBMC dataset, which is the real dataset with the most general mosaic scenario (Fig. 3a) used in this work. We ran scMoMaT under different hyper-parameter settings (similar to the settings used for the simulated data, with different values for d , λ , k and r), and summarized the result into barplots below.

From these results, we can see that scMoMaT is robust to the hyper-parameter setting of the regularization weight λ in the loss function, the number of neighbors k , and radius parameter r in the post-processing step. The performance of scMoMaT is mainly affected by the latent dimension d for cell and feature factors. A higher d helps to better separate cells of different cell types (higher ARI and NMI scores), though slightly reduces the alignment accuracy among batches (slightly lower GC score).

This pattern is consistent with observations from the results on simulated data (Fig. S17): the NMI and ARI scores stabilize once d reaches a certain threshold, which is a number of latent dimensions that is large enough for the given dataset. This threshold is smaller in simulated data than that in real data, which is expected, as d is associated with the complexity of the dataset and real data is often more complex than simulated data. The number of latent dimensions d is a parameter that exists in most methods for single cell omics data that involves dimensionality reduction, and it is commonly considered that a d that is too small can affect the performance.

We modified the main manuscript accordingly to include the test result on the human PBMC dataset (Lines 398-404, Page 14, and Fig. S18).

REVIEWERS' COMMENTS

Reviewer #2 (Remarks to the Author):

The authors' response has addressed all my questions.